# FEATURE SEGREGATION BY SIGNED WEIGHTS IN NEURAL NETWORKS FOR ARTIFICIAL AND BIOLOGICAL VISION

**Giordano Ramos-Traslosheros**
Department of Neurobiology
Harvard Medical School
Boston, MA 02115, USA
gio@hms.harvard.edu

**Carlos R. Ponce**
Department of Neurobiology
Harvard Medical School
Boston, MA 02115, USA
carlos_ponce@hms.harvard.edu

## ABSTRACT

Signed connectivity is fundamental to neural computation in both brains (excitatory/inhibitory) and machines (positive/negative). Yet the role of signed weights in shaping visual representations in object recognition remains unclear. Dale's Law, the biological principle that neurons send exclusively excitatory or inhibitory outputs, is typically not enforced in artificial neural networks (ANNs). Here, we find that accuracy in ImageNet-trained ANNs correlates with the spontaneous emergence of sign-specific "Dale-like" segregation in their output layers. Ablation and feature visualization reveal a functional segregation in ANNs: removing positive inputs primarily disrupts localized, object-related structure, while removing negative inputs alters mainly dispersed background textures. This segregation is more pronounced in adversarially robust models, persists with unsupervised learning, and vanishes with non-rectified activation functions. We validate these observations in the macaque ventral visual cortex (V1, V4, and IT) using encoding models and *in vivo* feature visualization. The features recovered by encoding models qualitatively matched those identified *in vivo*. Model representations changed more upon positive than negative input ablations. We analyzed the most Dale-like units across neuron models, positive units showed localized features, while negative units showed larger, more dispersed features. Consistent with this, experimentally clearing the background around a neuron's preferred feature enhanced its response, likely by reducing inhibitory drive. Our results suggest that both artificial and biological vision systems segregate features by weight sign: positive weights emphasize object-related features, while negative weights refine context. This highlights a convergent representational strategy in brains and machines, yielding predictions for visual neuroscience.

## 1 INTRODUCTION

Brains and artificial neural networks (ANNs) both rely on signed connections. In biological circuits, Dale's law states that neurons are either excitatory or inhibitory (Dale, 1935). Excitatory neurons are thought to compute core visual features, while inhibitory neurons sharpen selectivity, modulate context, and gate information flow (Isaacson & Scanziani, 2011). The primate ventral visual stream supports object recognition (DiCarlo et al., 2012), and representations from V1 to IT increase in complexity, paralleling hierarchies in convolutional neural networks (CNNs) (Yamins et al., 2014; Güçlü & van Gerven, 2017). While early mechanisms such as lateral inhibition and center-surround receptive fields are well characterized, the excitatory–inhibitory organization of higher ventral-stream areas remains less understood (Tamura et al., 2004), motivating the question of whether information is segregated by connection sign in both artificial and biological vision.

ANNs also use positive and negative weights, loosely analogous to excitatory and inhibitory signals but unconstrained by Dale's law. Yet it remains unclear how deep networks divide visual information across signed weights, particularly in object classification models where each output unit represents a category. Prior work suggests segregation by absolute weight strength (Li et al., 2023), but it is

unknown whether feature types such as foreground objects versus background context are systematically separated by weight sign. Here, we hypothesize that CNNs segregate visual information into positive and negative inputs.

We test this hypothesis across diverse ImageNet-trained CNNs by analyzing weight sign organization, ablating positive and negative inputs, and visualizing the resulting feature selectivity. We asses feature segregation across network depth by analyzing Dale-like channels. We examine a range of architectures and training regimes, including adversarially robust models, unsupervised models, and networks with nonrectified (Tanh) activations.

For biological comparison, we fit linear models from ANN features to neural responses recorded across the macaque ventral stream (V1, V4, IT) and use *in vivo* feature visualization and background manipulations to probe inhibitory contributions. These analyses relate ANN-derived sign structure to biological feature preferences.

Our results support an emerging principle: across artificial and biological vision systems, connection sign organizes feature representation. Positive weights emphasize object-related, localized features, whereas negative weights encode contextual, more dispersed structure. This connects classic ideas rooted in Dale's law with representational strategies in modern ANNs, generating mechanistic and testable predictions for visual neuroscience.

## 2 RELATED WORK

**Mechanistic interpretability of computer and biological vision** There has been progress in mechanistic interpretability in ANNs from work using perspectives adapted from neuroscience circuit dissection (Olah et al., 2020). This line of explainable AI research explains model behavior by dissecting smaller network subgraphs, revealing how relevant features arise from input weights and are composed hierarchically. Such work has uncovered motifs involving positive and negative connections between related features, reminiscent of early visual system organization. New approaches to address representations beyond single units rely on sparse dictionary learning, with early work in vision (Olshausen & Field, 1996), an approach that has recently regained popularity in language modeling (Cunningham et al., 2023), as well as in multimodal models (Pach et al., 2025). Some studies also characterized object shape and texture biases in feature visualizations by reconstructing images from sparse weight sets (Li et al., 2023). However, the systematic division between positive and negative inputs across the entire range of weight strengths, and its possible role in feature segregation, remains underexplored and is a focus of this study.

**Feature visualization by closed-loop optimization** Characterizing learned representations is foundational for both biological and artificial vision research. Feature visualization, i.e. optimizing for images that strongly activate target units, was originally pioneered in the brain by hand (Hubel & Wiesel, 1959), and later in silico by gradient ascent on pixels of neural networks (Erhan et al., 2009; Nguyen et al., 2016a;b; Olah et al., 2017). Because gradients are unavailable when recording in vivo, gradient-free black-box optimization techniques were developed for synthesizing preferred images of biological neurons in real-time (Ponce et al., 2019; Xiao & Kreiman, 2020; Wang & Ponce, 2022). These approaches constrain the search space via generative adversarial networks, promoting naturalistic solutions (Nguyen et al., 2016a). Further methods involve first fitting a predictive network to neural data and then using in silico gradient ascent (Bashivan et al., 2019; Walker et al., 2019). While most prior studies use grayscale images, our study applies gradient-free visualization to color images in both CNNs and primate recordings.

**Robustness** Neural networks are susceptible to adversarial attacks, where noise that is nearly imperceptible by humans can be added to natural images, changing output classification (Szegedy et al., 2014; Salman et al., 2020; Elsayed et al., 2018). Robust training, i.e. introducing adversarial perturbations during learning, improves resistance to such attacks and is hypothesized to align learned representations more closely with primate visual processing. Prior work does not assess how robustness impacts the organization of image representations after ablation of signed weights, which we systematically investigate here.

**Nonlinearity influence on representations** Beyond training objectives, the role of activation functions such as ReLU versus Tanh profoundly influences representational properties (Alleman et al., 2023), with ReLU inducing representations better aligned with input features and Tanh inducing

alignment with output features (labels). This prior work was done in small networks from a theoretical perspective; thus, the impact of rectification on the potential segregation of visual features at practical scales is unknown and addressed by our study.

## 3 METHODS

An extended methods section is in the Appendix A.1.

**Networks** We performed our ablation studies in CNNs pretrained on the ImageNet dataset: AlexNet (Krizhevsky et al., 2012), VGG16 (Simonyan & Zisserman, 2015), ResNet50 (He et al., 2015), and robust ResNet50 models with robustness radii specified by the $L_\infty$ norm ($L_\infty \in \{0.5, 1, 2, 4, 8\}$; Salman et al. 2020). Here, $L_\infty$ denotes the maximum-norm constraint used during adversarial training, i.e., $\|\delta\|_\infty \leq \epsilon$, which bounds the maximum per-pixel perturbation by $\epsilon$. To reduce computing time, we used the *imagenette* dataset (noa, 2024) and the *ImageNet* macaque category. We also tested 100 classes sampled by k-means on the output of ResNet50 (Figs. 12,13) For all networks, we visualized the representations of the units in the fully-connected output layer (pre-softmax) matching those classes under different ablation conditions.

**Dale index** We quantified Dale like structure with a Dale index for each outgoing channel per layer, defined as the fraction of its weights that share the majority sign, $D = \max(p_+, p_-)$, where $p_+$ and $p_-$ are the proportions of positive and negative outgoing weights. The index ranges from 0.5 to 1 and measures sign consistency.

**Ablation** For each layer, we ablated positive and negative weights separately. Given a layer's weight matrix $W$, we defined the sets of positive weights $P = \{w \in W : w > 0\}$ and negative weights $N = \{w \in W : w < 0\}$. For each set $S \in \{P, N\}$, we sorted its elements in decreasing order of absolute value. We then defined the ablation strength $\alpha \in [0, 1]$ as the fraction of the total magnitude of $S$ to remove. Specifically, we identified the smallest $k$ satisfying $\sum_{i=1}^{k} |w_i| \ / \ \sum_{w \in S} |w| \geq \alpha$, where $w_1, w_2, \ldots$ are the sorted weights in $S$, and set those $k$ weights to zero. Because $\alpha$ is a normalized cumulative magnitude, it lies in $[0, 1]$, and sweeping $\alpha$ from 0 to 1 removes none to all of the positive (or negative) weights.

**Feature visualization** For each ablation condition, we performed feature visualization by optimizing a GAN latent code to create an activity-maximizing image. We used this closed-loop, zeroth-order-search approach to allow comparison with our neuronal experiments, where gradient ascent would not be possible. To increase the span of the stimulus space, we used two GANs: AlexNet fc6 DeePSiM (Dosovitskiy & Brox, 2016) which can render textures and objects, and BigGAN (Brock et al., 2019) that can render photo-realistic images with objects. For optimization, we used a variant of *covariance matrix adaptation evolutionary strategy* or CMAES (Wang & Ponce, 2022; Loshchilov, 2015). We optimized ten images per GAN, resulting in 20 feature visualizations per output unit and ablation condition. Diverse visualizations better capture the multifaceted high-level representations in CNNs (Nguyen et al., 2016b). For our examples, we show the best of the 20 visualizations, but used all for quantitative analyses. For visualizations of neural networks predicting biological neuron responses, due to experimental time restrictions, we used five visualizations per ablation condition, via DeePSim only. Our experiments are performed in a PC with Nvidia 4090 GPU, and each visualization takes about 3 mins.

**Network training** Both ResNet18 networks were trained using the FFCV library (Leclerc et al., 2023) for 16 epochs on the ImageNet1K dataset. The top-5 classification accuracy was 0.797 for the network with Tanh activations and 0.870 for the network with ReLU activations. Note that these models were trained for only 16 epochs rather than the standard 90, so their accuracy underperforms published benchmarks. However, they are suitable for our mechanistic analyses.

**Visual cortex electrophysiology** We recorded multi-unit (neuron microcluster) and occasional single-unit activity from chronically implanted multielectrode arrays in V1, V4, and PIT of two macaques. Animals fixated while 2–8° images were flashed briefly (100 ms ON, 150 ms interstimulus interval). Neurons were driven with a 160-image stimulus set (diverseSet), constructed to span a broad range of visual features derived from k-means clustering of AlexNet output layer over the IN1K validation set and typical visual neuroscience image sets. For each session, we modeled responses of a single neuron or multiunit using a one-component PLS regression between firing rates

and AlexNet penultimate-layer activations, and used the resulting model for *in silico* ablations and feature visualizations, validating visualizations *in vivo* within the same session.

## 4 RESULTS

### 4.1 PROXIMITY TO DALE'S LAW IN CNN OUTPUTS CORRELATES WITH ACCURACY

A key challenge in comparing artificial and biological circuits is that CNNs are not constrained by Dale's law. Therefore, we asked whether the Dale index (our measure of sign consistency) in the output layer of diverse CNNs has any relation to their performance. Dale index increased from random initialization with training (Fig 1A). Moreover, top-1 accuracy on ImageNet1K positively correlated with the mean Dale index of the output layer. Within a given architecture, the Dale index increased with network depth. And batchnorm training in VGGs produced output layers with higher Dale index. Thus, even without an explicit Dale constraint, these networks naturally developed more sign consistent output channels. This motivated us to examine the specific visual features carried by positive versus negative weights.

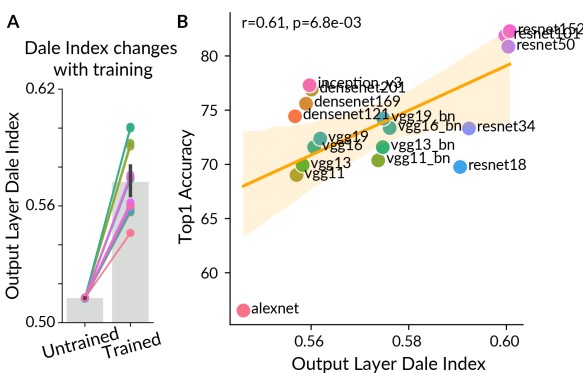

Figure 1: Dale index of output layer correlates with top 1 accuracy on ImageNet1K CNNs. **A.** Training increases Dale index of output layers. **B.** Network top 1 accuracy correlates with Dale index of the output layer.

### 4.2 OBJECT INFORMATION IS PREFERENTIALLY ENCODED BY POSITIVE WEIGHTS

**Hypothesis** Motivated by biological center-surround organization, where inhibitory surrounds convey contextual information, we hypothesized that CNN output units trained for object recognition segregate object features to positive weights and contextual features to negative weights.

**Testing segregation by ablation and visualization.** We examined this hypothesis in ImageNet-pretrained CNNs using ablation and feature visualization. The overall ratio of positive to negative input weights per unit was close to unity (Table 2), suggesting that both polarities may encode relevant information. We then selectively ablated positive or negative input weights to class units and visualized features across ablation strengths. Ablating positive weights greatly reduced the maximal achievable activation during feature visualization, whereas removing negative weights slightly increased it (appendix Fig. 11). Visually, positive-weight ablation disrupted recognizable object structure, while negative-weight ablation largely preserved object identity and instead altered background or color context (Figs. 2B, C). To quantify these effects, we compared image sets generated before and after ablation using mean pairwise cosine similarity across an ensemble of readout CNNs. Positive-weight ablation yielded substantially less similar representations, whereas negative-weight ablation produced only minor shifts (Fig. 2D). These results replicated across 100 ImageNet classes and with alternative metrics such as LPIPS (Zhang et al., 2018) (appendix Fig. 13), demonstrating robustness and generality.

To quantify to what extent objects disappear from the preferred images under ablations, we evaluated objectness using an object-detection network (YOLOv7 Wang et al. (2022)). Relative to baseline objectness scores computed from intact visualizations, ablating positive weights reduced objectness, whereas ablating negative weights had minimal impact (Fig. 2E). Analyzing spatial frequencies revealed positive ablations majorly affected low frequencies (app. Fig. 15,16), consistent with the objectness reduction. Together, in ImageNet-trained CNNs, removing positive input weights disrupts object features, while removing negative weights primarily alters context.

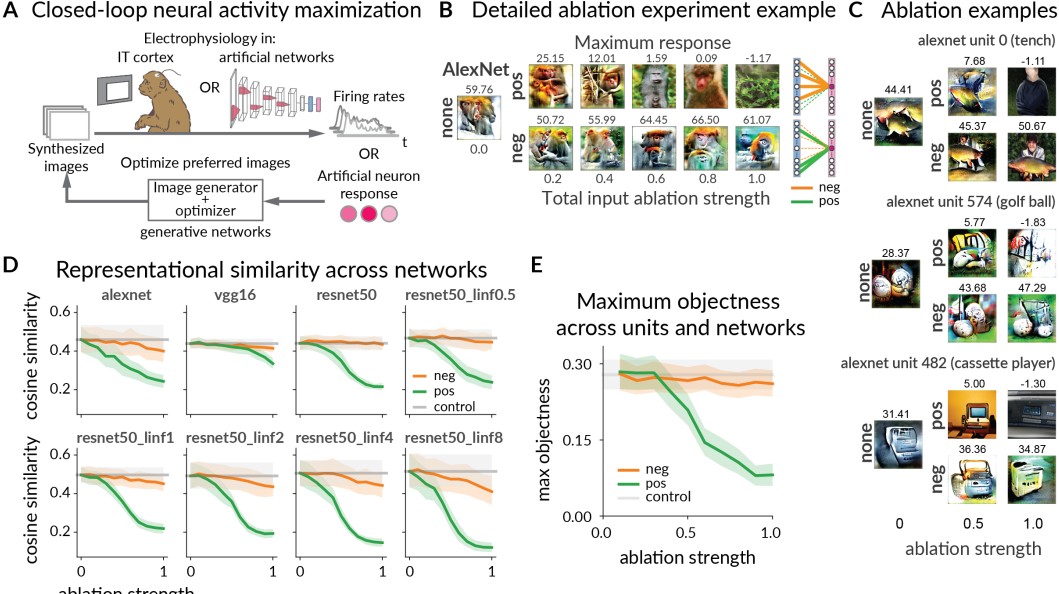

Figure 2: **A.** Schematic of feature visualization workflow in ANNs and brains. **B.** Preferred feature changes for different ablation strengths of input weights to the macaque 373 output unit of AlexNet (last fc layer of 1000 units before softmax). Images are the most activating images out of the 20 visualizations per ablation strength. Ablation strengths are below each image, and activation scores are above. **C.** Changes in preferred features to different ablations of example AlexNet output units: 0 tench, 574 golf ball, and 482 cassette player. Notice the large image changes for positive ablations. Same methods as in **B**. **D.** Representational similarity of intact vs input-ablated units across networks tested, measured by the pairwise cosine similarity of control vs ablation images over an ensemble of networks. Error bars are 95% confidence intervals over units, each unit is the mean of its 20 visualizations. The units correspond to the 10 imagenette categories ([0, 217, 482, 491, 497, 566, 569, 571, 574, 701]) plus the macaque category (373). **E.** Objectness scores across units per ablation condition. As in **D**, we tested 11 units from the 1000-unit fully-connected output layer (pre-softmax) of: AlexNet, VGG16, ResNet50, and robust ResNet50 ($L_\infty \in \{0.5, 1, 2, 4, 8\}$). For each network, we averaged over the objectness scores of 20 visualizations per unit and all units. The plot shows the mean over previously described network averages. Error bars are 95% confidence interval over network averages.

### 4.3 SEGREGATION DEPENDS ON RELU BUT NOT ON UNSUPERVISED PRETRAINING

We next explored potential mechanisms underlying this functional segregation by weight sign. To test segregation in unsupervised learning, we analyzed a ResNet50 backbone trained without supervision (SimSiam from Chen & He (2020)). After unsupervised pretraining, the backbone was frozen and a linear classifier was trained on ImageNet-1K. Feature ablation and visualization revealed that this network still allocated object features to positive weights, although these features disappeared at lower ablation strengths than in fully supervised CNNs (Fig. 3A, appendix Fig. 14). Negative input ablation had only a minor effect, suggesting that even unsupervised representations are organized so that positive weights convey most object-related features.

We hypothesized that this segregation depends on the rectifying ReLU activation, where the weight sign fully determines output sign, due to the non-negativity of activations. Prior work in toy networks shows that ReLU promotes alignment to the input space, whereas Tanh promotes alignment to the output space (Alleman et al., 2023). To test the role of rectification, we trained ResNet18 models with either ReLU or Tanh. As expected, the ReLU model showed clear segregation, with the strongest disruptions arising from ablation of positive weights. In contrast, the Tanh model showed similar representational changes across ablation types and preserved key features even when either sign was removed (Fig. 3B,C). Thus, rectified activations are required for strong segregation of object information into positive weights in CNNs.

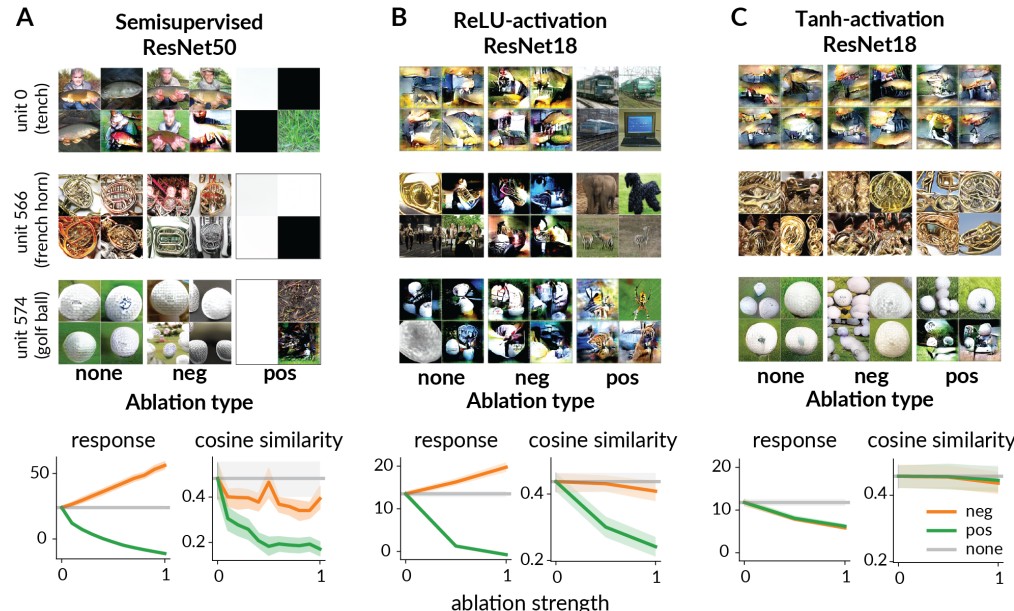

Figure 3: Ablation experiments (rows are features of three example units, and responses, and representational similarity for all 11 classes) in **A.** Semisupervised networks, **B.** Vanilla ReLU supervised network, and **C.** Network with a non-rectified activation function Tanh, replacing all ReLUs in **B**.

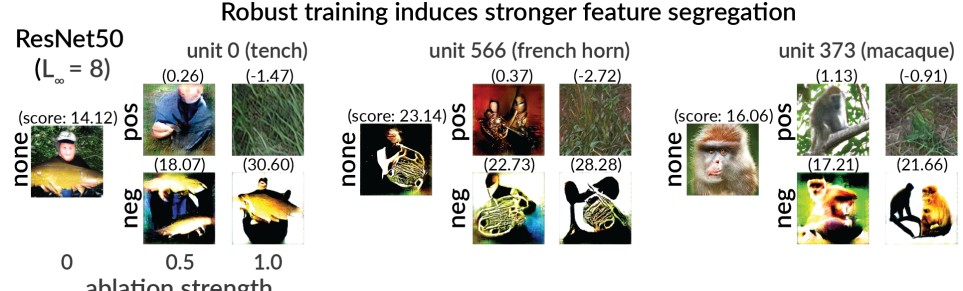

Figure 4: Robust network ResNet50 $L_\infty = 8$ shows a large change in preferred features upon input ablation. Notice the white background in the negative-weight ablation condition.

## 4.4 Adversarially robust networks show enhanced feature segregation

Given that unsupervised pretraining and rectification support this segregation, we next investigated adversarially robust networks. Adversarially robust networks, which are trained to resist small targeted image perturbations (Szegedy et al., 2014; Madry et al., 2019), are believed to better reflect aspects of biological vision and may therefore show distinctive patterns of feature segregation. We examined whether and how adversarial robustness influences the allocation of object and contextual information to positive and negative weights.

In robust ResNet50 networks, intact feature visualizations appeared more object-like, and ablation of negative input weights reliably altered the background color, often rendering it white (Fig. 4). This hinted at a stronger feature segregation than in vanilla networks. Quantitative analysis confirmed that as network robustness to adversarial attacks increased, so did the model's vulnerability to ablation, as measured by the difference in cosine similarity between control images and ablated images (see $\Delta$(cosine similarity) in Fig. 5). For ablation strength of 1 (yellow/light lines), the difference increased with network robustness, and slopes in Table 1 indicate that this trend holds across ablation polarities and strengths. Moreover, the robustness effects translated into higher shape bias in the benchmark by Geirhos et al. (2022). However, this benchmark did not disentangle ablation effects. Negative ablations similarly reduced shape and texture accuracy, preserving overall shape bias. Only robustness at $L_\infty = 4$ increased shape bias after ablating negative weights (Fig. **??**). Over-

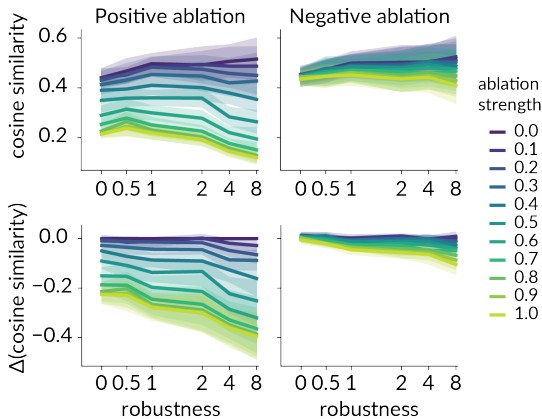

Table 1: Spearman correlation of representational change upon ablation vs robustness ($L_\infty$ norm)

| type | Positive | Negative |
|---|---|---|
| $\alpha$ | $\rho$ (pvalue) | $\rho$ (pvalue) |
| 0.1 | -0.17 (2e-1) | -0.10 (4e-1) |
| 0.2 | -0.39 (1e-3) | -0.21 (8e-2) |
| 0.3 | -0.34 (4e-3) | -0.14 (3e-1) |
| 0.4 | -0.38 (1e-3) | -0.34 (5e-3) |
| 0.5 | -0.47 (6e-5) | -0.46 (9e-5) |
| 0.6 | -0.48 (3e-5) | -0.34 (5e-3) |
| 0.7 | -0.51 (9e-6) | -0.52 (6e-6) |
| 0.8 | -0.50 (2e-5) | -0.49 (2e-5) |
| 0.9 | -0.48 (4e-5) | -0.62 (2e-8) |
| 1.0 | -0.47 (6e-5) | -0.57 (5e-7) |

Figure 5: Representational changes under input ablation increase with robust training in ResNet50. Top: cosine similarities to control. Bottom: changes relative to control. Most ablation strengths show significant correlations with robustness.

all, classification CNNs segregate object and context information to positive weights and texture or background information to negative weights, and adversarially robust training further sharpens this sign-based segregation.

## 4.5 FEATURE SEGREGATION IS NOT EXCLUSIVE TO CLASS UNITS

To determine whether feature segregation by weight sign is present beyond the output layer, we analyzed intermediate and early layers of CNNs. We first searched for channels that approximated Dale's law, by providing predominantly positive or negative inputs to their downstream units in each layer. We then visualized the preferred features of the most "Dale-like" channels using gradient-based methods with the Lucent library in PyTorch. Examining all five convolutional layers of AlexNet, we found that feature segregation by sign emerged throughout the network. In the first layer, channels with mostly positive weights responded to high-frequency achromatic edges, while those with mostly negative weights responded to lower-frequency, colored edges and spots. In the middle layers, positive channels tended to emphasize edges and detailed textures, whereas negative channels often represented broader, colored, or background-like patterns. By the final convolutional layer, channels with mostly negative weights produced features that resembled background elements such as sky or grass, while channels with positive weights highlighted sharp, localized object fragments like animal snouts and eyes (appendix Fig. 20). Altogether, our results show that feature segregation by weight sign is not restricted to the output layer, but gradually develops throughout the network. This pattern is reminiscent of Dale's law in biological circuits, suggesting that artificial neural networks can develop sign-consistent and functionally distinct representations across all layers, even in the absence of a biological constraint.

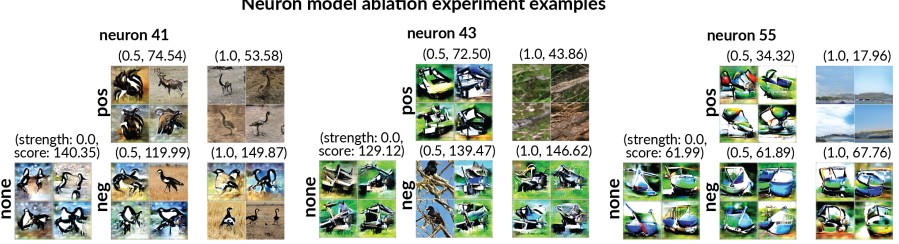

Figure 6: Preferred features of neuronal network models of visual neurons in the primate ventral stream. Pos: are positive ablations, neg are negative ablations, number indicates ablation strength. Shown are top 4 visualizations at 0, 0.5 and 1.0 ablation strenghts.

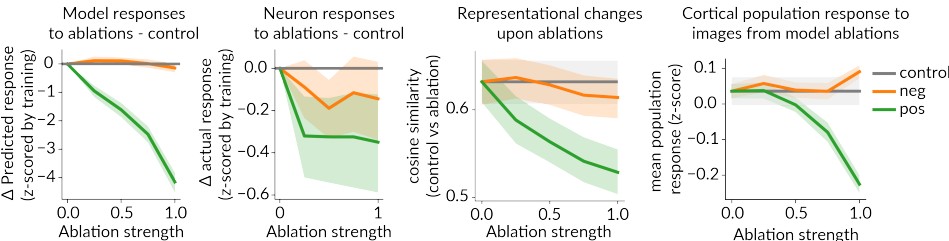

Figure 7: Left: predicted and actual neuron responses of model to ablations. Images obtained from positive ablations in the neuron models elicited a consistent activity drop on the biological neurons modeled. Right: Representational change of model to ablations measured by our cosine similarity metric on the neuron model feature visualizations upon ablation; and cortical population response to the images obtained from feature visualization from ablation of model units, neurons were z-scored before computing the population average. Plots show averages over 59 models, (35 for monkey C, and 24 for monkey D), shaded regions are the 95% C.I of the mean. For all plots the positive ablation condition was statistically different to the control.

## 4.6 BIOLOGICAL MODELS BASED ON IMAGENET NETWORKS SEGREGATE LOCAL FEATURE INFORMATION INTO POSITIVE WEIGHTS

The ventral stream in primates is responsible for object recognition, and artificial networks are the best models of its function. We therefore wondered if the segregation of positive and negative inputs we observed in networks might also occur in the brain. However, it is not currently possible to selectively remove positive or negative synaptic inputs from real neurons the way we can in artificial networks. To address this limitation, we fit linear models mapping CNN features to macaque ventral stream neural responses, and applied feature visualization to both model units and *in vivo* to biological neurons. This allowed us to test feature segregation in models of biological representations and to generate testable neuroscience predictions. We recorded neural activity from V1, V4, and IT cortex in two monkeys, using a diverse set of images and modeled each neuron's response using partial least squares regression with activations from the penultimate layer of AlexNet (4096 units). [1] We then applied our ablation and visualization protocol to these neuron models (see appendix for validation A.5) and showed the resulting images to the monkey during the same session. First, images from intact models reliably drove biological neurons to firing rates more than one standard deviation above those observed during natural image presentations (Fig.23, left), indicating out-of-distribution generalization. For the subset of neurons in which we performed in vivo closed-loop feature visualization, we found that the model's preferred features often matched those of the neuron, providing additional validation (appendix Fig. 23, right). However, in vivo features were more spatially localized (procedure in A.1), whereas in silico features exhibited greater spatial variation (rotated, mirrored, or repeated versions). This likely reflects invariances, due to using a fully connected layer, that are not present in our recorded neurons. Moreover, unlike the images from recognition units, images from the neuron models did not resemble objects (appendix Fig. 23, Fig. 6).

**Ablation experiments reveal sign-based segregation in neuron models.** Ablation experiments on these neuron models showed that removing positive input weights led to a significant decrease in both predicted and observed firing rates, while removing negative weights had a smaller effect (Fig. 7). This pattern was consistent across individual neurons and at the population level in the ventral stream (Fig.7, rightmost). While models are tested with out-of-distribution images produced by feature visualization, qualitative results hold for each sign ablation. Furthermore, changes in population activity by manipulation of single neuron models suggest that sign-based functional segregation in model predictions translates to measurable changes across the ventral stream population and perhaps perception.

---

[1]Although using other layers could improve predictive accuracy, we selected the penultimate layer to directly test if inputs optimized for classification maintain weight sign-based segregation in biological neural predictions.

**Dale's law inspired analysis.** To bring our models closer to Dale's law, we applied two approaches. First, we tested whether neural responses could be predicted using only positive input weights, corresponding to receiving input exclusively from excitatory artificial neurons. This constraint reduced both training and test accuracy relative to unconstrained models (appendix Fig. 24), indicating that neuron models require both positive and negative inputs. Second, we identified Dale-like artificial units that contributed mainly positive or negative weights to all output neuron models, defining putative excitatory and inhibitory inputs. Positive-weight units corresponded to smaller-scale edges and localized spots, cleanly separated from the background, whereas negative-weight units aligned with broader textures and larger patches (appendix Fig. 25). This pattern supports the idea that inhibitory-like artificial inputs preferentially encode contextual or background structure, paralleling inhibitory modulation in biological cortex.

**Experimental manipulation of background as a test for inhibition.** To further test this hypothesis, we experimentally manipulated image backgrounds *in vivo*. In a subset of experiments, we presented altered images in which the background was cleared around the neuron's preferred feature, thereby reducing the putative inhibitory drive. As predicted, this manipulation resulted in increased neuronal responses (appendix Fig. 26), providing functional support for the idea that inhibitory or negative inputs are involved in contextual modulation and that their reduction can enhance feature selectivity in high-level visual cortex. Together, these results suggest that functional segregation by input sign extends to models of ventral stream neurons, providing concrete testable predictions for future experiments targeting excitation and inhibition in visual cortex.

## 5 LIMITATIONS

Our results hold in the last layer units of multiple networks. Due to limited computing time, we did not test all 1000 categories in as many networks as possible, our largest test consisted of 100 units. While larger scale simulations will provide exhaustive evidence, we are confident our main claims will stand. We limited our neuron recordings to a 160 image dataset for regressing neuron responses via CNNs. While we observed good fits and recovered relevant feature to the neurons, more images may improve the models, especially using larger-scale versions of our diverseSet. The neuroscience results would need to follow Dale's law to be mapped one-to-one to excitatory and inhibitory neurons, but we make no claim to a perfect mapping in this work. We observed a range of sign-segregation levels across architectures and depths, thus we expect a gradient in the functional segregation across architectures and training regimes. Such an example is our finding of increasing robustness training leading to stronger segregation, but we anticipate more. The fundamental question of shape vs texture, foreground vs background remains to be solved. We reconciled changes in frequency structure with changes in objectness and visual representations and LPIPS image similarity. However, more work remains to understand the full extent of the segregation reported here. Solving this problem for the visual cortex may provide better benchmarks for this task in AI.

## 6 DISCUSSION

Our study combined ablations with feature visualization guided by naturalistic image priors to reveal the functional segregation of class-level features in the output layer of ImageNet trained CNNs: positive weights contribute object/shape/low frequency information, while negative weights contribute background/contextual/texture information. This effect was enhanced in robust networks, it was present in networks with unsupervised pretraining, but was absent in network trained with Tanh instead of ReLU. Our results explain how the background contribution to classification observed in (Xiao et al., 2020) emerges, backgrounds are primarily encoded by the negative inputs.

Importantly for neuroscience, the observed functional segregation in neuron model units in CNNs hints at a functional segregation in the brain beyond the center-surround classically studied in V1. We also crafted a diverse dataset for visual neuroscience recordings that is scalable. Neuron responses to a smaller but diverse set of naturalistic, colored images, with complex foregrounds and backgrounds, led to models capturing relevant features obtained experimentally from the neuron. Thus, using both model-based and model-free approaches revealed richer neuronal representations. Preferred images from neuron models with positive input ablations elicited smaller average population responses of cortical neurons. This suggests that ablation in networks modeling neurons

holds potential as a method to control the population activity in the brain. To relate ablation-induced changes in the images to the population responses is a future direction. This ablation based on the natural division of positive and negative weights can be easily extended into arbitrary layers, e.g., using gradients to define positive and negative contributions to any arbitrary unit. Our ablation approach proposes baselines for the functional differences between excitatory and inhibitory neurons in higher cortical visual areas. The functional segregation has consequences for neural coding and response selectivity. Our findings generate concrete predictions for future experiments using advanced genetic or optogenetic tools to dissect excitation and inhibition in primate cortex. Understanding the circuit mechanism of biological vision could aid further understanding and development of computer vision models. Ultimately, these insights suggest that biological constraints like Dale's Law emerge from functional demands. Our work thus introduces consistency in signed connectivity as a biologically grounded primitive for the growing toolkit of mechanistic interpretability.

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

AUTHOR CONTRIBUTIONS

G.R.T. led the conceptualization, methodology, experiments, analysis, and manuscript writing. C.R.P. supervised the project and contributed to reviewing and editing the manuscript.

ACKNOWLEDGMENTS

We thank Elizabeth Cleaveland for their technical help with data acquisition and surgical procedures. This research was supported by the Alice and Joseph Brooks Fellowship (G.R.T.), the William Randolph Hearst Fellowship (G.R.T.), the Common Fund of the National Institutes of Health under award 1DP2EY035176-01 (C.R.P.), and the David and Lucile Packard Foundation (2020-71377, C.R.P.).

# A APPENDIX

## A.1 EXTENDED METHODS

**Networks** The ablation studies were performed on CNNs pretrained on the ImageNet dataset: AlexNet (Krizhevsky et al., 2012), VGG16 (Simonyan & Zisserman, 2015), ResNet50 (He et al., 2015), and robustly-trained ResNet50 ($L_\infty \in \{0.5, 1, 2, 4, 8\}$, Salman et al. (2020)). All these networks end on a 1000-unit fully connected layer, each unit corresponding to one of the 1000 ImageNet categories. Neural networks were used in Pytorch.

**ImageNet subsampling** To reduce computing time, for most of the experiments, we used a subset of ImageNet, the *imagenette* dataset (noa, 2024) and the macaque category, 11 classes in total. These classes and their corresponding output units in each network trained on the 1000-class ImageNet dataset are as follows: (0, tench), (207, English Springer), (482, cassette player), (491, chain saw), (566, church), (569, French horn), (571, garbage truck), (574, gas pump), (701, golf ball), (970, parachute), and (373, macaque). We visualized the representations of the output layer units of those classes under different ablation conditions. For Fig. 12, to sample 100 diverse classes out of the 1000 ImageNet classes, the 50k validation images were first clustered into 100 clusters via agglomerative clustering of the L2 distance matrix from the 1000-d output features of ResNet50, which was pre-trained on ImageNet. Then, one new unique class is selected from each cluster.

**Ablation** We used two ablation conditions: we ablated weights that were (1) only positive or (2) only negative. We ablated weights cumulatively by first sorting the positive (or negative) weights by their (absolute) decreasing value. We defined the *ablation strength*, $\alpha$, as a fraction of the total positive or total negative weights to a unit. We identified the top $k$ weights necessary to reach the silencing strength, i.e., $\sum_{i=1}^{k} w_i \leq \alpha$, and set them to zero. We covered the range of ablations from 0 to 1. For most experiments with ANNs, we used silencing strengths in 0.1 steps, from 0 (intact) to 1 (complete ablation).

## Closed-loop neural activity maximization

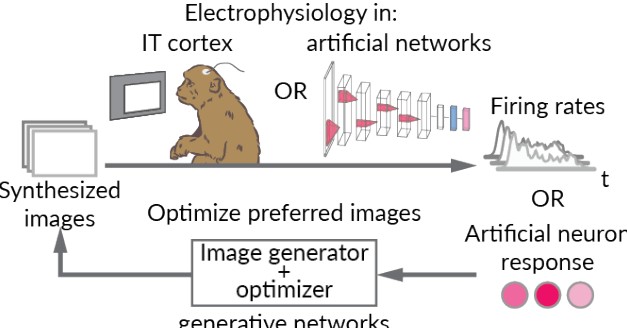

Figure 8: Schematic of feature visualization workflow in ANNs and brains. Optimizer is CMAES, image generators are DeePSim fc6 or BigGAN.

**Feature visualization** For each ablation condition, we performed feature visualization by optimizing a GAN latent code to create an activity-maximizing image Fig. 8. We used this closed-loop, zeroth-order-search approach to allow comparison with our neuronal experiments, where gradient ascent would not be possible. To increase the span of the stimulus space, we used two GANs: AlexNet fc6 DeePSiM (Dosovitskiy & Brox, 2016) and BigGAN (Brock et al., 2019). For optimization, we used a variant of *covariance matrix adaptation evolutionary strategy* or CMAES (Wang & Ponce, 2022; Loshchilov, 2015). Initial conditions for the CMAES were given as standard deviation of 3.0 for DeePSim, and 0.2 for BigGAN. Initial images for the algorithm were small norm vectors for both GANs, close to the origin of the latent spaces. For BigGAN, we generated a fixed noise vector by scaling a 128-dimensional truncated noise sample (-1.4, 1.4), and concatenated it with a 128-dimensional zero vector of the class embedding, to form the required 256-dimensional input code. The remaining parameters are determined by the dimensionality of the search space of each

AlexNet output
feature space (PCA)

160DiverseSet

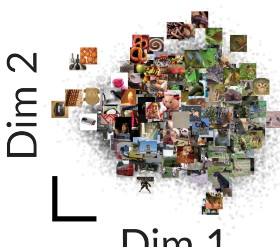

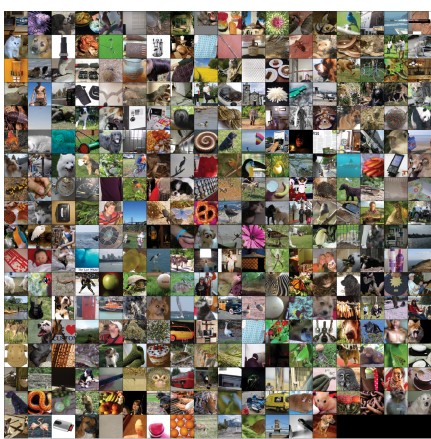

Figure 9: Illustration of a diverse dataset construction using AlexNet output feature space. The embedding is the output of the last layer before softmax of AlexNet, a vector space of 1000-dimensions. Left: PCA showing the coverage of the feature space by the diverseSet 160, only for illustration purposes. Right: images from diverseSet 160 used to fit neuron models.

GAN. We optimized ten images per GAN, resulting in 20 feature visualizations per output unit and ablation condition. Diverse visualizations better capture the multifaceted high-level representations in CNNs (Nguyen et al., 2016b). For our examples, we show the best of the 20 visualizations, but used all for quantitative analyses. For visualizations of neural networks predicting biological neuron responses, due to experimental time restrictions, we used five visualizations per ablation condition, via DeePSim only. Our experiments are performed in a PC with Nvidia 4090 GPU, and each visualization running 100 iterations takes about 3 mins. For *in vivo* experiments, we ran from 20 to 60 iterations of the AlexNet fc6 DeePSiM with the CMAES algorithm implemented in Matlab, linked to our real-time spike-sorting data acquisition. The responses fed to the CMAES algorithm were the average firing rate on the window 70-170 ms from image onset.

**Feature analysis**    We computed image similarity using an ensemble of CNNs, including AlexNet, ResNet50, and ResNet50 with robustness in $L_\infty \in \{0.5, 1, 2, 4, 8\}$, inspired by (Feather et al., 2023) And confirmed the results with LPIPS (Zhang et al., 2018) in the appendix. We computed their activations and defined similarity as the average pairwise cosine similarity (LPIPS) between control activity vs input-ablated activity. We averaged the results of the CNNs ensemble, resulting in one quantity per ablation condition. We computed *objectness* as the maximum bounding box score provided by YOLOv7 (Wang et al., 2022), this was averaged over visualizations per unit, units per network, and then across networks.

**Visual cortex electrophysiology**    We collected data from two animals (monkey C and monkey D), each implanted chronically with floating multielectrode arrays (Microprobes for Life Sciences, MD) of 32 or 16 channels (monkey C, N = 96 electrodes, monkey D, 64), in areas V1, V4 and posterior inferotemporal cortex (PIT). All institutional procedures were followed. Channels were distributed as (V1, V4, PIT): monkey C (32, 32, 32), monkey D (16, 16, 32). Some electrodes captured the activity of single units, but most showed multi-unit activity (reflecting the pooled activity of micro-clusters of neurons). The animals performed a simple fixation task, which required them to keep their eyes on a 0.25-deg diameter spot at the center of the screen, within a square fixation window measuring 0.5–1° per side. Images were presented for 100 milliseconds ON, 150-ms off, 4-5 images per trial, after which the animal received water or juice. Images were presented to monkey C were 2 deg in size, and 4-8 deg for monkey D to match the receptive field centers of most channels in all cortical areas (V1, V4 and PIT). Image presentation and data acquisition (electrophysiology, eye tracking) were integrated by the MonkeyLogic2 software (Hwang et al., 2019) and OmniPlex Neural Recording Data Acquisition Systems (Plexon Inc.), interfaced through custom Matlab code. We performed online spike sorting using the PlexControl client based on waveforms. We used ViewPixx

EEG monitors (ViewPixx Technologies), at a resolution of 1920x1080 pixels with 120 Hz refresh rate. Eye tracking used ISCAN cameras (ISCAN Inc.). And reward was delivered using the DARIS Control Module System (Crist Instruments).

**Feature localization in vivo**   We conducted a perturbation-based localization to identify relevant image regions from a feature visualization performed in vivo, where gradient information from the animal brain is unavailable. We perturbed a circular region with a 50-pixel diameter within the 256-pixel image by randomly shuffling the pixels inside this circle, effectively disrupting the local image structure while maintaining local contrast. We selected 30 such regions for perturbation at random, excluding those that extended beyond the image boundaries. The modified images were then presented to the monkey. We hypothesized that perturbing regions crucial for driving the neuron response would lead to a decreased firing rate. To assess local image importance, we calculated the normalized response change: the difference between the firing rate response to the intact image and the firing rate response to the perturbed image, divided by the firing rate response to the intact image. A normalized response change of 0.5 indicates the neuron response decreased by half due to perturbation. To generate the localized response mask, we averaged the circular masks corresponding to each perturbed region, weighted by their response change. This response mask was further smoothed using a Gaussian kernel with a 30-pixel standard deviation. We defined relevant regions as those causing a normalized response change of 0.5 or greater. Finally, we applied this mask to the original feature visualization image to highlight the local features.

**Image dataset**   We collected a reference image dataset to activate neurons in the monkey along the hierarchy of V1, V4, and PIT. Because neurons vary in their preferred features, we constructed a dataset spanning the image space as represented by the neural embedding of ImageNet-trained AlexNet. The embedding is the output of the last layer before softmax of AlexNet, a vector space of 1000-dimensions. The images from this dataset also spanned uniformly the 1000-dimensional output space of a semi-supervised trained network, trained on a billion images, ResNet50SS (Yalniz et al., 2019). To define this embedding space, we performed PCA on the output activations from AlexNet to the 50k ImageNet validation images, we kept the top 300 components (accounting for about 95% of total explained variance). Then we partitioned the space into a defined number of clusters $k$, according to the desired dataset size, using batched k-means to reduce computational burden. After finding the $k$ cluster centers, we could feed arbitrary images to the network, map them to the PCA space, and then pick the nearest neighbors to the cluster centers from the desired image space. In addition to the ImageNet validation set, we added other common neuroscience datasets (Brady et al., 2008; Kar et al., 2019; Allen et al., 2022; Hung et al., 2005) to form our image space. We selected $k = 160$ images, as a set that was diverse but small enough to be used in every experimental session. We called this image dataset *diverseSet* .

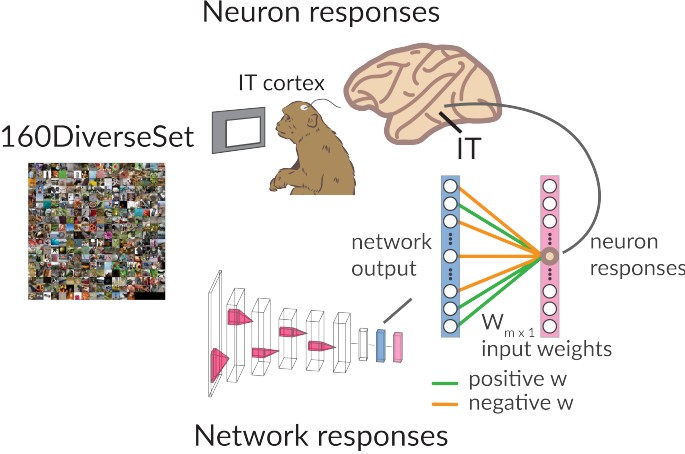

Figure 10: Schematic of model fitting using the dataset diverseSet. 160 images were split into train/test datasets (80/20).

**Models fit on neuronal activity** We recorded responses of neurons in the ventral stream to a 160 image dataset, our diverseSet Fig. 10. We relied on a small dataset to fit neuron responses and perform feature visualizations within the same experimental session. We performed partial least-squares linear (PLS) regression (80/20 train/test split) between the neuron responses to images and the activations of the penultimate layer of AlexNet. We used one component for the PLS regression. We selected one neuron or microcluster per experimental session, fitted a model, and performed the ablation and feature visualizations *in silico* for that model. We selected the best fitted neuron per session, based on the $r^2$ on the 20 % held out test set, usually in the range of 0.15 to 0.5. When time allowed, we also performed the feature visualization of the modeled neuron *in vivo* using a gradient-free approach (Ponce et al., 2019), within the same experimental session. To test whether features learned by the model were relevant to the biological neuron, we recorded the neuronal responses to the preferred images of the model. We then analyzed the representational similarity of the model features under ablations using ANNs. And analyzed the responses of the biological neuron populations from V1, V4 and IT.

## A.2 SUPPORTING RESULTS ON NETWORK ABLATIONS

Table 2: Ratio of positive to negative weights. We divided the sum of positive weights by the sum of the absolute values of the negative weights.

| MODEL | RATIO (MEAN $\pm$ STD) |
|---|---|
| AlexNet | $1.03 \pm 0.08$ |
| VGG16 | $1.01 \pm 0.09$ |
| ResNet50 | $1.00 \pm 0.06$ |
| ResNet50 ($L_\infty = 0.5$) | $1.00 \pm 0.05$ |
| ResNet50 ($L_\infty = 1$) | $0.99 \pm 0.05$ |
| ResNet50 ($L_\infty = 2$) | $1.00 \pm 0.04$ |
| ResNet50 ($L_\infty = 4$) | $1.00 \pm 0.05$ |
| ResNet50 ($L_\infty = 8$) | $1.01 \pm 0.05$ |

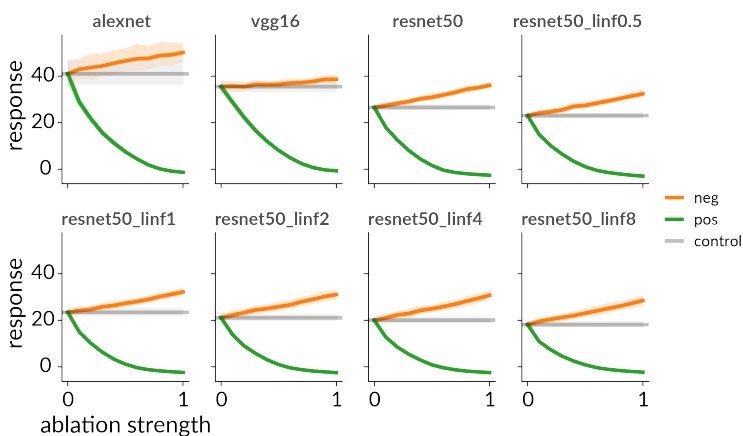

Figure 11: Mean activation scores of units used in ablation experiments. For all networks, units scores come from the last fully-connected layer, with 1000 units, before the softmax. The units correspond to the 10 imagenette categories ([0, 217, 482, 491, 497, 566, 569, 571, 574, 701]) plus the macaque category (373). Error bars are 95% confidence intervals over units (categories tested), where each unit response is the mean of its 20 visualizations. *Control* refers to the feature visualizations in the intact networks for the same units, we extended it as a horizontal line to ease visual comparisons to the different ablation strengths.

**ResNet50 last fc layer units: 10x larger dataset, 100 new imagenet classes**

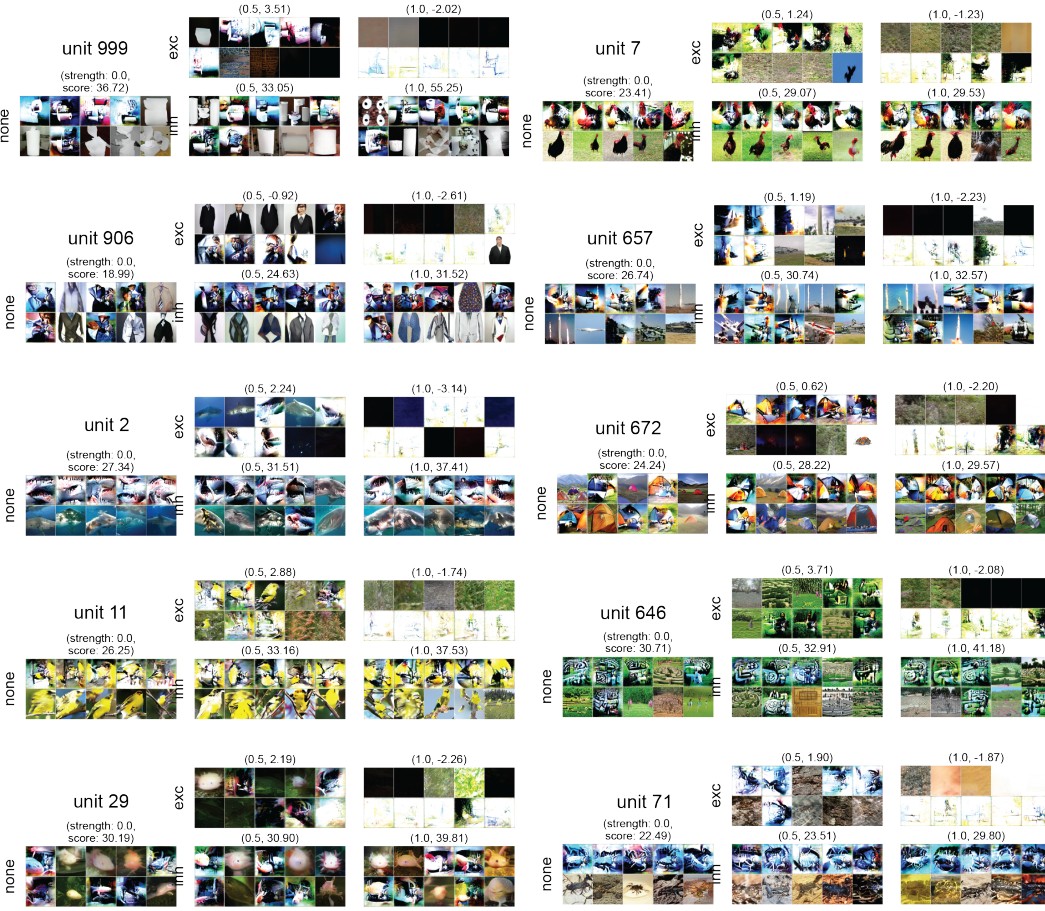

Figure 12: Functional segregation holds in a 10x larger dataset. 100 classes out of the 1000 ImageNet categories were selected by clustering the 50k validation images embedded in the 1000-d output space of ResNet50 picking one class per cluster. Thus, we now have 10x more data points that should span the representational space of the output layer we study. Consistent with the smaller dataset, the main object features degrade into more uniform background images upon positive ablation. Here we show examples from 10 of the 100 classes.

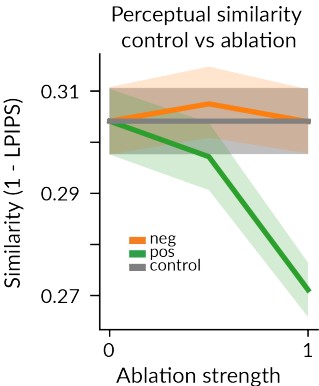

Figure 13: Functional segregation holds in a 10x larger dataset with LPIPS (Zhang et al., 2018) as representational similarity measure. We measured the representational similarity of the images as 1 - LPIPS among control images and between control images and ablation images (Fig. 12). We average results per class, and show the mean and 95% C.I. across the 100 classes. The representational similarity degrades upon positive input ablations, confirming results obtained from the imagenette dataset.

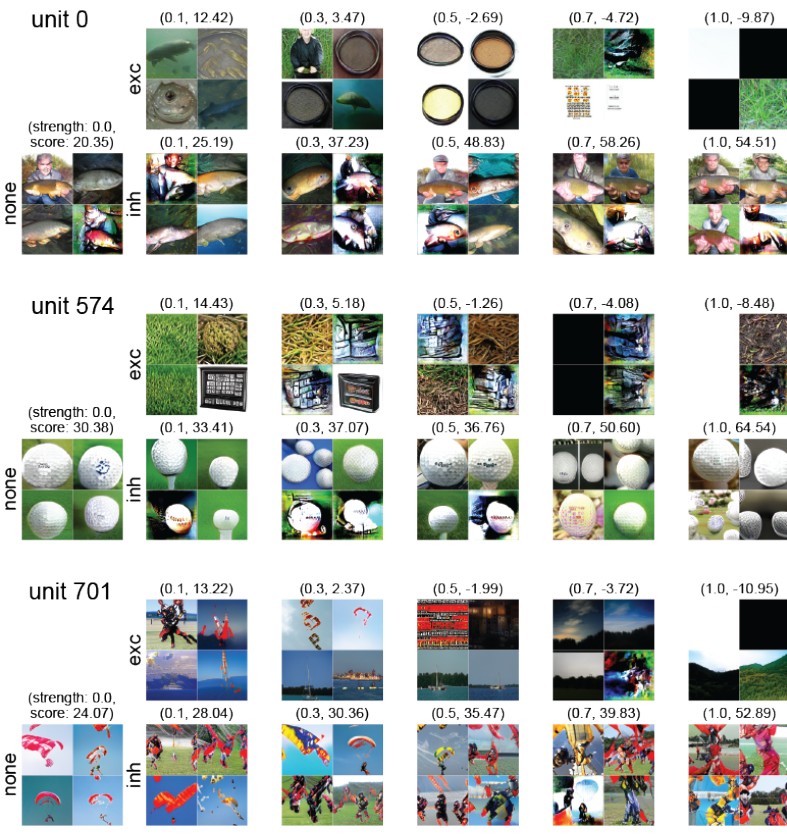

Figure 14: Feature visualizations of ablation experiments in a network pretrained with unsupervised learning. ResNet50SimSiam (Chen & He, 2020). The unsupervised network with frozen weights was coupled to a fully connected layer, only this layer was fine-tuned to classify ImageNet1000. Network units changed starting with small positive weight ablations, see unit 574 golf ball. Smaller changes are visible upon negative weight ablations, however object relevant features remain. Overall behavior is consistent with CNNs trained directly on ImageNet1000 classification.

### A.3 SPATIAL FREQUENCY CONTENT, SHAPE/TEXTURE, AND OBJECT/BACKGROUND ANALYZES

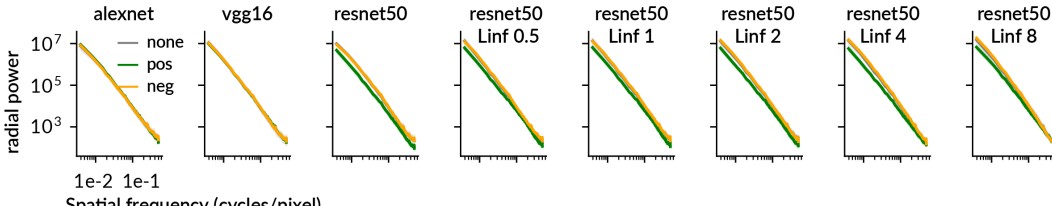

Figure 15: Functional segregation of frequency content. We measured the radial power spectrum for control and ablation images of different networks. We average results per class, and show the mean and 95% C.I. across the 11 classes. Low frequencies degrade upon positive input ablations, while negative ablations overlap with control spectra.

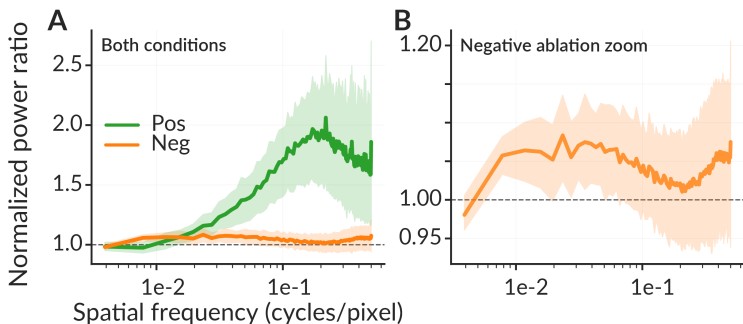

Figure 16: Functional segregation of frequency content. We measured the radial power spectrum for control and ablation images of different networks. We average results per condition and per network, and show the ratio of control to ablated spectra. The mean and 95% C.I. are across networks. Low frequencies degrade upon positive input ablations, while negative ablations overlap with control spectra but slightly enhance low frequencies.

We measured the accuracy on the background challenge (Xiao et al., 2020). To define reliance on background and foreground we used the `bg_only_t` and `fg_only` conditions, $R_{bg}$ and $R_{fg}$ respectively. We normalized differences of baseline accuracies using the `original` condition of the intact networks (Fig. 19).

For background reliance, the control condition showed mean $R_{bg} = 0.157 \pm 0.116$ (s.d.), which increased to $0.488 \pm 0.631$ under positive weight ablation (mean change $\Delta R_{bg} = +0.331$; Wilcoxon signed-rank test vs. control, $p = 0.027$) and decreased to $0.078 \pm 0.046$ under negative ablation ($\Delta R_{bg} = -0.079$; $p = 0.0078$). Although networks cannot classify above chance solely using backgrounds, the performance they can achieve can be more strongly attributed to negative weights. This limitation supports the approach of feature visualization vs a benchmark that was used retrain networks rather than testing pretrained ones. For foreground reliance, the control condition had mean $R_{fg} = 0.903 \pm 0.326$, changing to $0.839 \pm 0.777$ under positive ablation ($\Delta R_{fg} = -0.064$; $p = 0.359$) and to $0.711 \pm 0.245$ under negative ablation ($\Delta R_{fg} = -0.193$; $p = 0.0117$). However, these results are biased by the accuracies under chance Fig. 18 causing outliers in the robust networks. Networks with above chance accuracy in the control conditions, such as ResNet50, AlexNet and VGG16, show decreased performance upon positive ablations (neg only, green) Fig. 18, (pos only means negative ablation). Taken together, with some caveats, positive weights reduced background reliance and increased foreground reliance in this experiment.

### A.4 DALE'S LAW INSPIRED ANALYSIS OF WEIGHTS AND INTERMEDIATE FEATURES

We showed Dale index (DI) increased from random initialization to training Table 3. To determine whether weight segregation of features occurs beyond the output layer, we visualized feature representations that predominantly provide negative or positive inputs to subsequent layers in AlexNet.

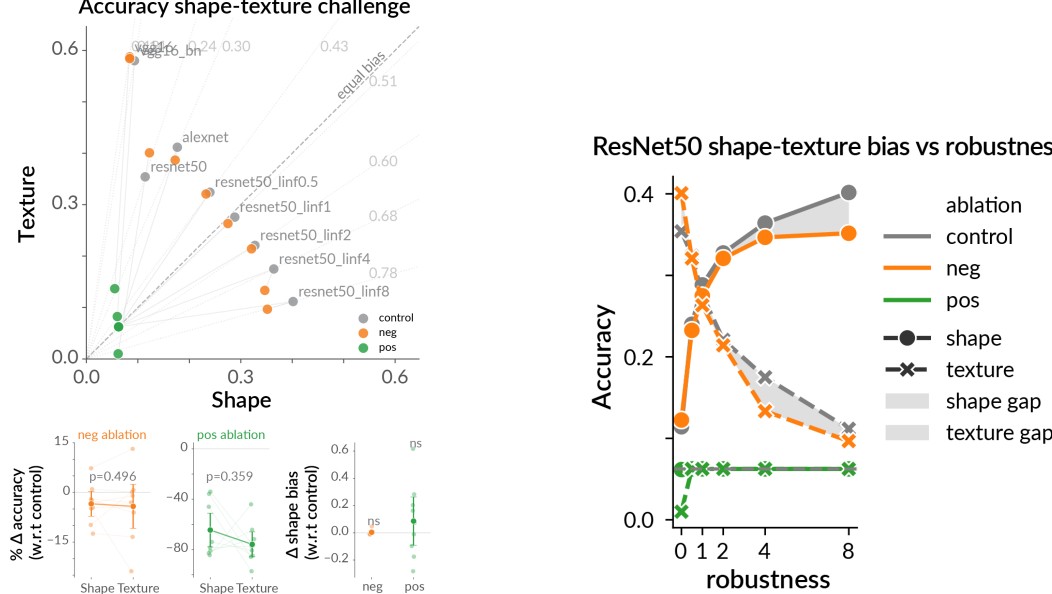

Figure 17: Functional segregation of shape vs texture using ablations in the shape/texture bias benchmark (Geirhos et al., 2022). (Left) Effect of ablations on shape vs texture encoding across our networks. (Right) Effect of robustness on shape vs texture encoding in ResNet50. While robustness shifts networks to be more shape-biased, this benchmark is affected equally by both ablation types.

Table 3: Dale index of the final classification layer before and after training, and trained Top1 accuracy.

| Model | Untrained DI | Trained DI | $\Delta$ DI | Top1 Acc |
|---|---|---|---|---|
| alexnet | 0.5126 | 0.5461 | +0.0335 | 56.52 |
| densenet121 | 0.5125 | 0.5567 | +0.0442 | 74.43 |
| densenet169 | 0.5126 | 0.5590 | +0.0464 | 75.60 |
| densenet201 | 0.5126 | 0.5601 | +0.0475 | 76.90 |
| resnet18 | 0.5128 | 0.5905 | +0.0778 | 69.76 |
| resnet34 | 0.5130 | 0.5923 | +0.0793 | 73.31 |
| resnet50 | 0.5125 | 0.6004 | +0.0879 | 80.86 |
| resnet101 | 0.5126 | 0.5998 | +0.0872 | 81.89 |
| resnet152 | 0.5127 | 0.6007 | +0.0880 | 82.28 |
| vgg11 | 0.5123 | 0.5570 | +0.0447 | 69.02 |
| vgg11_bn | 0.5123 | 0.5737 | +0.0614 | 70.37 |
| vgg13 | 0.5125 | 0.5583 | +0.0457 | 69.93 |
| vgg13_bn | 0.5125 | 0.5747 | +0.0622 | 71.59 |
| vgg16 | 0.5126 | 0.5606 | +0.0480 | 71.59 |
| vgg16_bn | 0.5126 | 0.5761 | +0.0635 | 73.36 |
| vgg19 | 0.5129 | 0.5619 | +0.0490 | 72.38 |
| vgg19_bn | 0.5129 | 0.5748 | +0.0619 | 74.22 |
| inception_v3 | 0.5127 | 0.5597 | +0.0470 | 77.29 |

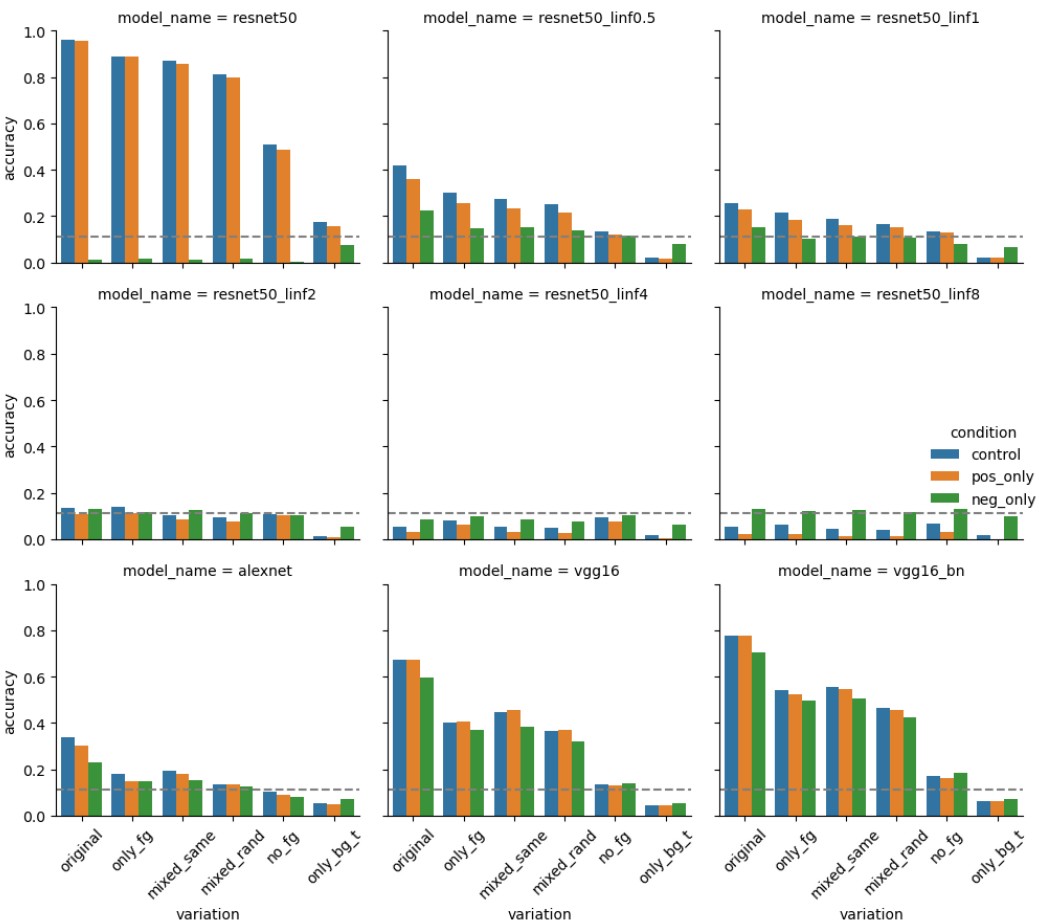

Figure 18: Functional segregation of background vs foreground. We measured the accuracy on the background challenge (Xiao et al., 2020). Robustness increases the role of negative weights in encoding backgrounds but below the chance level. Counterintuitive results show either sign can solve this challenge in AlexNet and VGG.

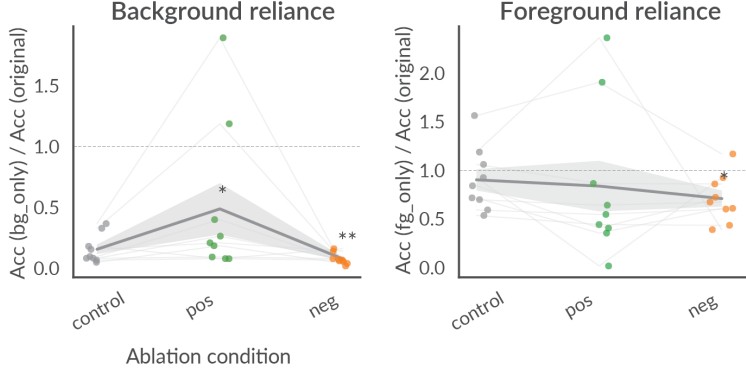

Figure 19: Functional segregation of background vs foreground. We measured the accuracy on the background challenge (Xiao et al., 2020).

We calculated sign consistency by averaging spatial weights and determining the frequency of positive and negative weights across output channels. The visualization of sign-consistent input features was conducted using the Lucent library in PyTorch, leveraging gradient-descent channel activity maximization. We focus on AlexNet's intermediate layers, examining the top and bottom sign-consistent features for each input channel.

Layer Details:

Conv1: Conv2d(3, 64, kernel_size=(11, 11), stride=(4, 4), padding=(2, 2))
Conv2: Conv2d(64, 192, kernel_size=(5, 5), stride=(1, 1), padding=(2, 2))
Conv3: Conv2d(192, 384, kernel_size=(3, 3), stride=(1, 1), padding=(1, 1))
Conv4: Conv2d(384, 256, kernel_size=(3, 3), stride=(1, 1), padding=(1, 1))
Conv5: Conv2d(256, 256, kernel_size=(3, 3), stride=(1, 1), padding=(1, 1))

Features that contributed mostly positive weights differed from the features contributing mainly negative weights, with object vs background arising with increasing depth. This positive vs negative weight split is evident even in the first layer, where low-frequency color features are contrasted with high-frequency black-and-white features Fig. 20.

## A.5 BIOLOGICAL NEURON MODELS AND EXPERIMENTAL VALIDATION

For each recording session, we selected the best model for further analysis, based on predictive accuracy (mean test $r^2 = 0.27 \pm 0.10$ SD across sessions). The fitted models included both positive and negative input weights, with a mean ratio of 1.17 for the sum of positive to negative absolute weights (Fig. 22). Our final dataset comprised (V1, V4, pIT): (7, 5, 23) neurons in Monkey C and (1, 5, 18) in Monkey D, with the majority of data from pIT cortex.

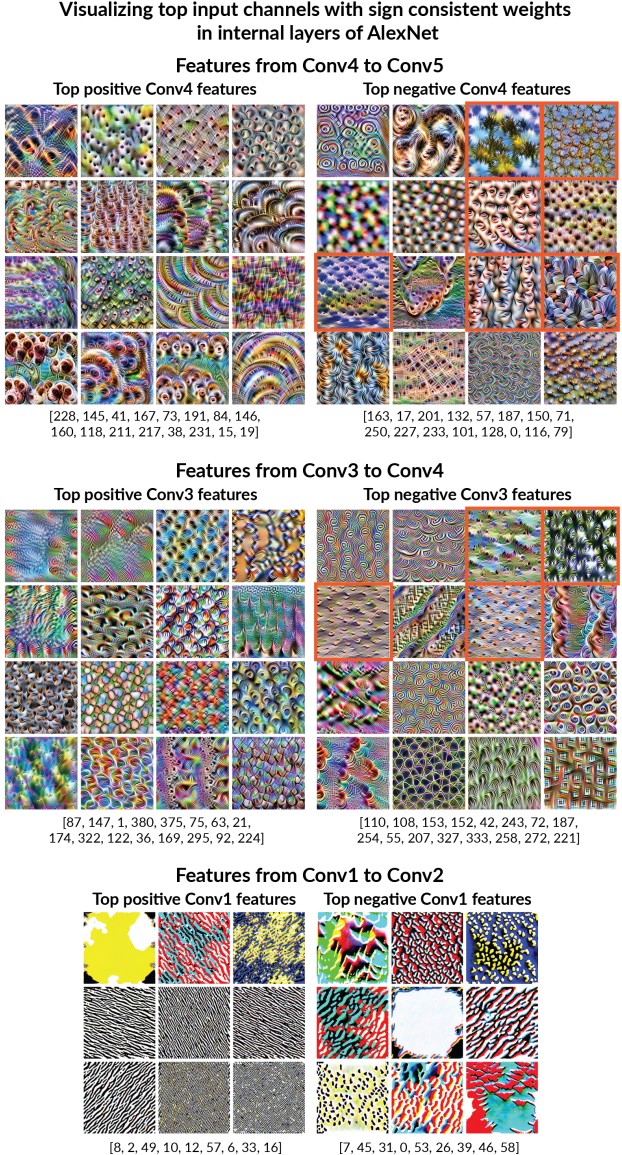

Figure 20: Layer Conv5 from Conv4: Features contributing mainly negative weights resemble backgrounds, such as patches of sky and grass, and sometimes face-like features (e.g., in the tench class), highlighted in orange borders. Positive weights align with localized object-like fragments, such as snouts and eyes of animals, and sharp spotted textures vs the blurry spotted textures for negative weights. Layer Conv4 from Conv3: Negative features still incorporate some background elements like ground or grass textures (orange borders), together with some spiral, square and blurry textures. Positive features exhibit more heterogeneous textures and higher frequency details, without evident background-like textures. Layer Conv2 from Conv1: Positive weights carry high-frequency edges mostly without color, while negative weights include lower frequency edge features and spotted textures with color, overall more spatially coarse. Channel index from the visualized features is shown as a list below each panel.

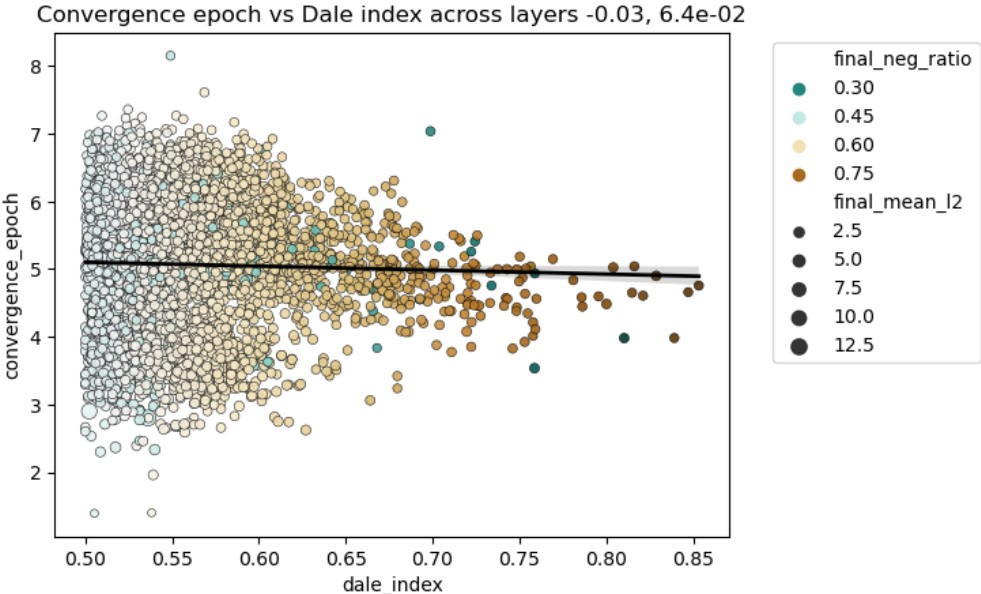

Figure 21: Weight dynamics vs Dale index. Convergence rate of outgoing weights measured by cosine similarity to final weights, as time to reach 90% of the final similarity. Each dot is one output channel for all layers in ResNet18 trained over 16 epochs. Color indicates proportion of negative signs. There is no obvious correlation. However, high Dale index channels mature within narrower time windows than more mixed channels.

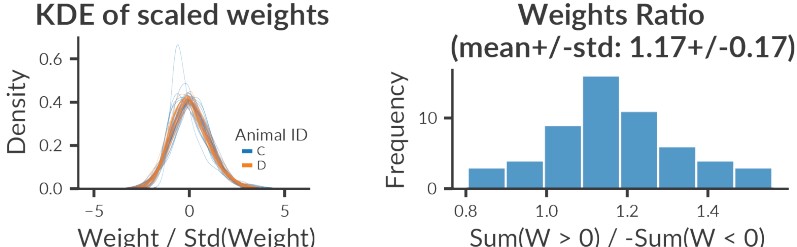

Figure 22: Left: Distribution of the model weights from neuronal fits with AlexNet penultimate layer features. Each model maps 4096 parameters from penultimate layer of AlexNet to the response of one biological neuron. Models use positive and negative weights. Model weights were normalized by their standard deviation to plot them on the same scale, for sake of visualization. Right: Ratio of total positive to total negative weights, per neuron model. Models use slightly larger positive weights with a mean of 1.17 and std of 0.17. Model numbers: 35 for monkey C, and 24 for monkey D.

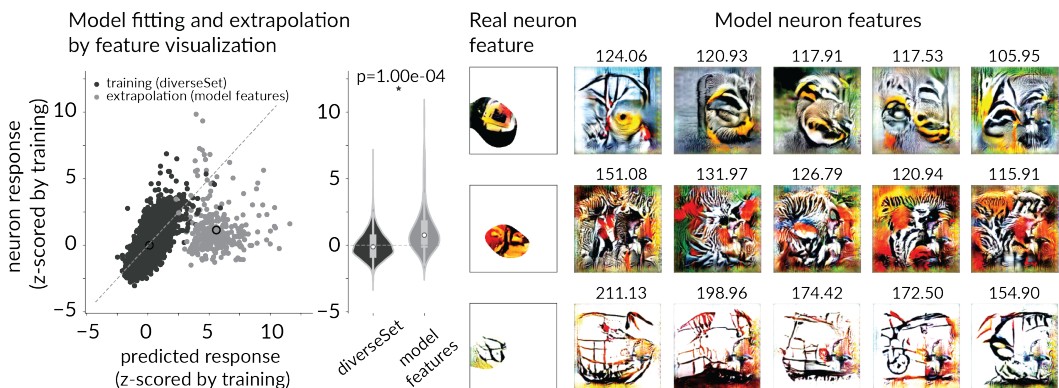

Figure 23: Neuron model units recover features relevant for the biological neurons. Left: Responses vs predicted responses of neurons to the training images, and the extrapolated features visualized from the intact models, which are extrapolations because the training data did not cover those high response ranges. Permutation t-test of neuron responses shows higher responses to images from model features than the natural images of the training dataset (diverseSet). Right: three neuron examples that show the feature visualization of the preferred feature of the neuron masked by the full-width at half-maximum obtained from perturbations to the image, and to their right the five feature visualizations of the intact model with the real neuron responses to those images on top.

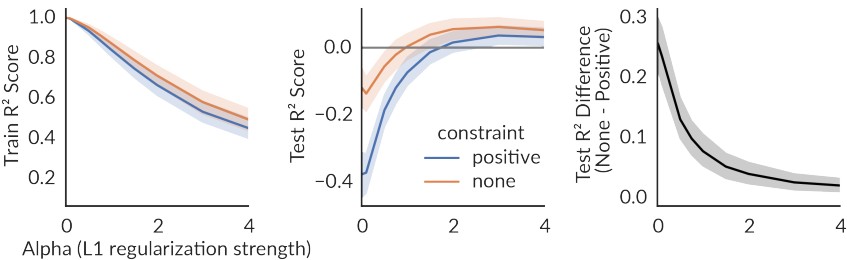

Figure 24: Using negative weights improves neuron models obtained via Lasso regression. Lasso regression models were fit with and without the positive constraint, over a 5-fold cross validation. Models were a linear regression from the 4096 features to a single neuron, over all neurons modeled from both animals. Left: performance on the training set measured by $r^2$ score. Middle: $r^2$ performance on the test set. Right: Model improvement by using positive and negative weights vs using only positive weights given by the difference in $r^2$ on the test set. Unconstrained models perform better than the positively constrained model, across the range of L1 penalties (sparseness penalty) tested, suggesting negative inputs from artificial network features are useful to predict biological neuron responses.

**Features from neuron models, AlexNet 4096 ReLU fc layer**

Positively weighted    Negatively weighted for >90% neuron models

Figure 25: Features that had positive or negative weights in most of the neurons models ( 91% of the 56 neurons, binomial test $p = 5.09e^{-10}$). These features are the closest approximation to features respecting Dale's law from our models. Left: best of 20 feature visualizations for the features with positive weights across neurons, feature index is on top of the image. Features are from the penultimate fc layer post ReLU, containing 4096 units. Right: best feature visualization from the negatively weighted features across neurons. Positively weighted features contain more local features like curved edges, while negative features contain textures or larger image patches. Sign consistency tested for statistical significance against the Bernoulli distribution of 0.5 probability with Bonferroni correction for testing 4096 features.

Background clearing can enhance neuron responses to the original feature visualization

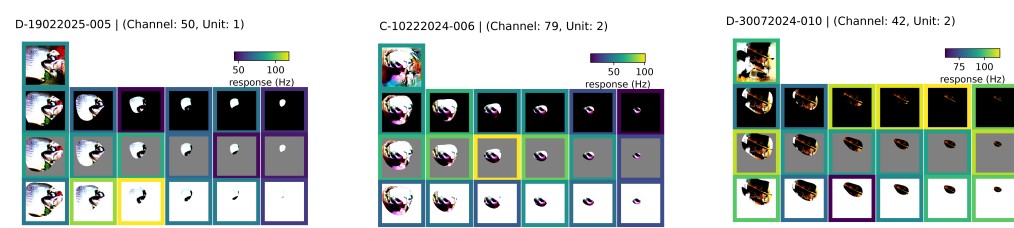

Figure 26: Clearing the background around the images obtained via closed-loop visualization can further boost responses in real-time recordings. Examples of 3 neurons in 2 monkeys.

