# OpenReview forum: "Feature segregation by signed weights in artificial vision systems and biological models"
_ICLR.cc/2026/Conference — ICLR 2026 Poster_

### Official Review · Reviewer_sJXU · 2025-10-23

**Soundness:** 2
**Presentation:** 2
**Contribution:** 2
**Rating:** 2
**Confidence:** 3

**Summary:**

In this paper, the authors study the role of positive and negative weights in artificial neural networks and their analogues with excitatory and inhibitory synapses in biological neural networks.  By performing visualization and ablation experiments in a variety of convolutional neural networks, the authors suggest that positive weight connections primarily encode object-relevant features while negative weight connects play a larger role in encoding background content.  The authors further train linear mappings to predict biological neural activity in monkeys from learned representations in CNNs.  Performing further visualization and ablation experiments with these learned mappings, the authors show the impact of projection weight sign on biological neural firing rate and conclude that both biological and artificial visual systems segregate features based on weight sign.

**Strengths:**

- The authors motivate their work based on biological principles.  Studying feature segregation in the visual system with image models is an interesting idea and mechanistic understandings that result from such analyses can be important to better understand information encoding in both biological and artificial systems.
- Analyses provided in this paper extend beyond the study of artificial neural units.  The authors sought to corroborate their findings in-vivo.

**Weaknesses:**

- With regard to sensitivity of output unit weights to positive and negative connections (results section 4.1), why would we expect anything different than the results presented (i.e., that positive output weights are important for encoding object-relevant information and negative weights for non-object information)?  Since these connections directly contribute to an output unit that is trained to have high activation only when a specific object is present (via the classification loss), shouldn't we expect these observed results as our default hypothesis?  And if so, why would this not also explain why we saw a similar, but lesser effect, when studying unsupervised models and no segregation in models trained with tanh units (which could contribute evidence for the presence of an object to an output unit by multiplying a negative activation with a negative weight or positive activation with a positive weight)?
- I am unconvinced by the qualitative analysis of section 4.4.  In early and mid layers, there is no evidence that visualized positive (negative) features are used primarily for object (background) information.  All visualized features in these early layers could likely be activated with various objects (or background textures).  More causal and quantitative evidence is needed to support this claim.
- Analyses in section 4.5 rely on the assumption that the model predicts biological neural activity well on stimuli outside of the training set.  This is claimed to be true (line 411 “images from intact models reliably drove biological neurons to firing rates…indicating out-of-distribution generalization”) but Figure 16, left, would suggest otherwise: outside of the training data, neuron responses appear to be poorly predicted by the model (what are the $r^2$ scores for held out in-distribution predictivity and out-of-distribution predictivity?).  Driving biological neurons to fire more than they do for natural images could simply be a result of the fact that the presented synthetic images are out of the natural image distribution.

**Questions:**

- For analyses in section 4.5, why were units from monkey visual areas V1 and V2 aligned with the penultimate representations in AlexNet when earlier layers of CNNs tend to be better predictors of activity in the early ventral stream?
- With regard to weakness bullet point 3, what is the variance explained (or pearson correlation) between biological neural activity and model neural activity for extrapolated images?

---

> ### Author Response · Authors · 2025-12-03
> **Reply to review**
>
> > With regard to sensitivity of output unit weights to positive and negative connections (results section 4.1), why would we expect anything different than the results presented (i.e., that positive output weights are important for encoding object-relevant information and negative weights for non-object information)? Since these connections directly contribute to an output unit that is trained to have high activation only when a specific object is present (via the classification loss), shouldn't we expect these observed results as our default hypothesis? And if so, why would this not also explain why we saw a similar, but lesser effect, when studying unsupervised models and no segregation in models trained with tanh units (which could contribute evidence for the presence of an object to an output unit by multiplying a negative activation with a negative weight or positive activation with a positive weight)?
>
> - We agree with the logic of the reviewer. However, it is not trivial to assume one could make a feature in a unique way. For example, selectivity for a doughnut could be achieved by two positive half-doughnut inputs or by a positive large circle input and a negative input suppressing a smaller inner circle.
> - Weight segregation is indeed enforced by ReLUs, which is why we have a loose analogy to biological neurons which are also thresholded below some input level.
> - Since shape/texture interact with object and background interpretations, we find the distinction is not trivial. And show that for robust ResNet50 negative weights contribute more to texture than shape bias, but still contribute to both (Fig. 16).
>
> > I am unconvinced by the qualitative analysis of section 4.4. In early and mid layers, there is no evidence that visualized positive (negative) features are used primarily for object (background) information. All visualized features in these early layers could likely be activated with various objects (or background textures). More causal and quantitative evidence is needed to support this claim.
>
> - We added new quantification of spatial frequency contributions by weight sign in Fig. 14, 15. We can accept a relaxation to interpret our reported functional segregation by a frequency rather than object information. However, our new results show that robust training enhances this segregation making a clearer case for our original statement in terms of object or shape vs background or texture (Fig. 16).
>
> > Analyses in section 4.5 rely on the assumption that the model predicts biological neural activity well on stimuli outside of the training set. This is claimed to be true (line 411 “images from intact models reliably drove biological neurons to firing rates…indicating out-of-distribution generalization”) but Figure 16, left, would suggest otherwise: outside of the training data, neuron responses appear to be poorly predicted by the model (what are the scores for held out in-distribution predictivity and out-of-distribution predictivity?). Driving biological neurons to fire more than they do for natural images could simply be a result of the fact that the presented synthetic images are out of the natural image distribution.
>
> - We need extensive animal time to validate our results in vivo. We believe we recapitulated relevant features obtained via feature visualization in vivo with our models (Fig. 21), and that perturbing such models nudges the neuron responses in the expected direction even if not in absolute magnitude (Fig. 7). Thus, generating higher responses with images that contain features similar to those obtained directly from the neuron in closed-loop shows the model is useful out-of-distribution. The firing increase is over 1 std of the training set and not orders of magnitude higher.
> - Moreover, we have a model-free experiment that shows that manipulating feature surround/context/background can increase neuron responses (Fig. 24.). This is consistent with a release of inhibition by removing suboptimal features surrounding the main excitatory input feature.
> - Most of the neurons are from V4 and IT, so the penalty of fitting V1 neurons with late layers is minimal.

---

### Official Review · Reviewer_udxK · 2025-10-31

**Soundness:** 2
**Presentation:** 3
**Contribution:** 3
**Rating:** 6
**Confidence:** 4

**Summary:**

This paper investigates an interesting organizational principle in neural networks: whether positive and negative weights segregate different types of visual information. Through systematic ablations and feature visualizations across multiple ImageNet-trained architectures, the authors demonstrate that positive weights preferentially encode localized, object-like features while negative weights encode more dispersed, texture-like or background features. They show this segregation is enhanced in adversarially robust models, persists with unsupervised pretraining, but critically depends on ReLU activations. The authors extend their investigation to primate ventral stream recordings, fitting linear models from CNN features to neural responses and performing both in silico and in vivo feature visualizations.

**Strengths:**

## Strengths

* Originality: The systematic investigation of feature segregation by weight sign is a novel angle in interpretability research. While prior work has explored weight pruning and sparsity, the specific hypothesis that signed weights might organize visual information differently (analogous to biological E/I circuits) is creative and underexplored. The connection to adversarial robustness is particularly original.


* Quality: The experimental design is thorough and rigorous. Testing across multiple architectures (AlexNet, VGG16, ResNet50, robust ResNets) with different training regimes (supervised, unsupervised, robust) provides strong evidence that this is a general phenomenon rather than an architectural artifact. The ablation methodology is sound, using cumulative weight removal based on magnitude. The use of two different GANs (DeePSiM and BigGAN) for feature visualization helps ensure results aren't generator-specific. Scaling to 100 ImageNet classes (Fig. 11) and validating with LPIPS demonstrates robustness of findings.


* Clarity: The paper is generally well-written with effective visualizations. Figure 1 provides a clear overview of the phenomenon. The comparison between ReLU and Tanh networks (Fig. 2) elegantly isolates the role of rectification. The progression from output layers to intermediate layers (Section 4.4) is logical. Methods are sufficiently detailed for reproduction.

* Significance: This work has potential impact for both AI interpretability and neuroscience. For AI, it offers a new lens for understanding how networks organize information and suggests that analyzing positive/negative pathways separately could aid interpretability. The connection to adversarial robustness is important - if robust models show stronger segregation, this could inform development of more interpretable models. For neuroscience, while the biological validation is preliminary, the approach of using ablation-based feature visualization to generate predictions for circuit experiments is valuable and could inspire new experimental designs.

The finding that ReLU is necessary for segregation connects nicely to recent work on how activation functions shape representational geometry, adding to our theoretical understanding of deep learning.

**Weaknesses:**

## Weaknesses

While the core ANN experiments are well-executed and reproducible, several issues limit the soundness of the overall claims:
* **Interpretation ambiguity:** The central interpretation of "object vs. background" segregation is not sufficiently justified. The observed effects could equally reflect:

1. Local vs. global spatial structure
2. High vs. low spatial frequency content
3. Shape vs. texture information
4. Figure vs. ground organization

The YOLOv7 objectness metric provides only weak support, as it reflects one specific computational definition of "object" that may not align with what the networks actually learned. I recommend: (1) testing alternative interpretations using texture/shape metrics (e.g., Geirhos et al. 2019 style analyses), (2) frequency domain analysis of the features, and (3) more direct tests of foreground/background using segmentation masks.

* Biological validation concerns. The neuroscience component has significant limitations:

1. R² = 0.27 is quite low for claiming the models "capture" neural representations
2. 160 images are substantially a small amount (although I believe experiments could be tricky, I'm afraid this could limit the final conclusions)
3. The leap from positive/negative ANN weights to excitatory/inhibitory neurons oversimplifies Dale's law, which involves distinct cell types, complex dynamics, and circuit-level interactions not present in ANNs
4. The penultimate layer may not be optimal for predicting ventral stream responses

I suggest: (1) comparing against larger image sets to assess whether findings hold, if possible, (2) testing multiple layers to find optimal predictions, (3) being more careful about the analogy to E/I balance, and (4) acknowledging these are computational models of neurons, not direct biological measurements.

Lastly, I believe that the paper excellently demonstrates **what** happens but provides no insight into **why**. What properties of the learning objective, training dynamics, or network architecture cause this segregation to emerge? Adding:

1. Analysis of weight evolution during training
2. Theoretical framework or toy models
3. Predictions about when this would/wouldn't occur would significantly strengthen the contribution.

-----

The paper is generally well-written with clear figures. However, some improvements would help:

* The "Dale's law inspired analysis" in Section 4.4 and Appendix A.3 feels somewhat disconnected from the main narrative. Consider integrating it more smoothly or clarifying its relationship to the classification unit findings.
* Figure 1B-C would benefit from showing more than just the best visualization per condition to convey variability
* The biological methods (Section A.1) could be condensed, with some details moved to supplementary materials
* Some key results (like the 100-class validation) are relegated to appendix; consider moving to main text

**Questions:**

## Questions for Authors

To summarize the points raised above:

* Alternative interpretations: Have you tested whether the segregation reflects spatial frequency rather than object/background? Could you show power spectra of features preferred by positive vs. negative weights?

* Training dynamics: When does this segregation emerge during training? Does it correlate with the development of robust features or adversarial robustness?

* Mechanistic predictions: Can you predict which architectures or training procedures would show stronger/weaker segregation based on some principle?

* Causality: The ablation experiments show correlation, but do they demonstrate that positive weights cause object representations? Could there be confounding factors?

### Minor Issues

* Table 2 shows positive/negative weight ratios are very close to 1:1. This is interesting but underexplored. Why this balance?
* The limitations section is quite brief; consider expanding

---

> ### Author Response · Authors · 2025-12-03
> **Reply to reviewer suggestions**
>
> Thank you for highlighting these strengths. We appreciate your recognition of the novelty of our weight‑sign–based analysis, the rigor and breadth of our experiments, and the clarity of our presentation. We’re also glad that you found the interpretability, robustness, and neuroscience implications meaningful.
>
> We are also grateful for critics and suggestions to strengthen our results.
>
> ### Weakness 1 (object interpretation)
> The following suggestions were addressed in new analyses and figures
> - Local vs. global spatial structure (Fig. 14, 15)
> - High vs. low spatial frequency content (Fig. 14, 15)
> - Shape vs. texture information (Fig. 16)
> - Figure vs. ground organization (Fig. 17)
>
> From these results we can correlate low frequency with shape and object encoding. Although some networks would conflate shape and texture according to Geirhos 2019, the main contribution to their bias would still be from positive weights. Clear interpretation is the frequency biases of each weight sign, with positive ablations affecting mainly low frequency content. Interestingly, we bring new perspectives into robustness interpretability showing robust training increases the shape bias as expected from our feature visualizations in Fig. 4. Prior work on figure vs background trained networks to solve those challenges, instead we analyzed pretrained performance changes under input ablation and see that negative weights contribute to foreground and background but not in a clear cut way (Fig. 17). From these, we conclude that objectness and spatial frequency are the most robust interpretations of the functional weight segregation.
>
> ### Weakness 2 (biological validation concerns)
> We validate our model perturbations in vivo, and see they can recover features present in the in vivo feature visualization, and that models can also steer responses in the expected direction (Fig. 7) although with less fine control than in ANNs. Experimental validation is done within session, which includes closed-loop feature visualization, so using larger image sets is not feasible for this design. Nevertheless, we have model-free evidence that modulating the background around the feature causes response increases, and this could be compatible with release of inhibition elicited by suboptimal contexts.
> We are more explicit in out limitations to the biological analogies from our models, but again provide extra model-free evidence in Fig. 24.
>
> ### Weakness 3 (Training dynamics)
> > Training dynamics: When does this segregation emerge during training? Does it correlate with the development of robust features or adversarial robustness?
> - We have not managed to train new robust models for this rebuttal to address dynamics. When we train a vanilla ResNet18 we see no correlation in how long a channel takes to become stable in learning versus the final Dale index it achieves (Fig. 25). However, high Dale index channels mature within narrower time windows than more mixed channels.
>
> ### Weakness 4 (mechanistic predictions)
> > Mechanistic predictions: Can you predict which architectures or training procedures would show stronger/weaker segregation based on some principle?
> - We show in Fig1. that VGG with batchnorm has higher Dale-index in their output layer, and that Dale index increases with network depth within given architectures.
>
> #### Minor points
> We did not delve deeper into the balanced weight ratio, which is indeed puzzling. We expanded limitations section to address challenge of background and foreground vs shape texture as confounding factors. We show several visualizations per condition across most figures, but have tight space in Fig. 2 (previously Fig. 1).

---

### Official Review · Reviewer_kwmx · 2025-11-01

**Soundness:** 2
**Presentation:** 1
**Contribution:** 3
**Rating:** 4
**Confidence:** 4

**Summary:**

This paper is a commendable piece of interdisciplinary research that bridges mechanistic interpretability in deep learning with fundamental principles of neurobiology. The overall contribution is substantial, and the authors' efforts to validate their computational claims with biological data are notable. However, the work suffers from weaknesses in its presentation, methodological rigor, and conceptual clarity across both the AI and Neuroscience domains.

**Strengths:**

1) The core contribution is the novel insight that the **signed weights ($+/-$) in ANNs serve distinct, functionally segregated roles**, which moves beyond treating weights merely as mathematical optimization parameters. This work is a crucial step toward understanding the computational significance of weight structure.

*   **Interpretability Breakthrough:** The paper provides a clear, interpretable meaning for the sign of weights—positive weights encode **object features**, and negative weights encode **background/contextual information** (i.e., feature segregation). This is a significant advance in explaining *why* modern deep networks achieve robust visual representations.
*   **A Solid Link to Neuroscience:** The work addresses a fundamental question in both fields by investigating whether a balanced mixture of positive and negative signals is required for representations. This alignment of **computational principles (signed weights)** with **neurobiological function (excitatory/inhibitory balance)** makes it particularly relevant for interdisciplinary venues like ICLR.

2) The methodology for cross-validation is robust, demonstrating a deep commitment to scientific rigor and biological plausibility.

*   **Biological Plausibility:** The use of **novel, complex V4 neuronal recordings from macaque monkeys** is a major strength. This direct, first-hand biological validation significantly elevates the paper's credibility, ensuring the claimed segregation is not a mere artifact of the ANN architecture but a potentially universal principle of visual processing.
*   **Effective Computational Methodology:** The authors employed an appropriate and sophisticated methodology—**customized activation maximization (feature visualization)**—to probe the specific functional preference of both positive-only and negative-only weighted inputs. This technique is well-suited for visualizing the intrinsic features encoded by the network, moving beyond simple classification tasks.
*   **Architecture and Stimuli Relevance:** The choice of **ResNet** architecture is appropriate given its deep usage in vision and common comparison with the visual processing stream, enhancing the generalizability of the findings.

3) The paper's execution required non-trivial effort across multiple domains, reflecting the significant endeavor of the research team. Conducting novel electrophysiology experiments and integrating them with advanced computational analysis is a demanding task that is not commonly accomplished, warranting praise for the authors' hard work.

**Weaknesses:**

1) The paper suffers from a general lack of clarity and transparency that severely impedes the reviewer's ability to assess its claims and ensures reproducibility.

*   **Insufficient Referencing and Background:** The Introduction lacks essential references to support foundational claims and necessary background knowledge, particularly in the neuroscience domain. This poor referencing undermines the academic rigor required to justify the "fundamental question" being investigated and the paper's overall scholarly context. While some of references are addressed in related works, referencing only 2 papers in 6 paragraphs of introduction needs to be extensively revised considering the majority of the community is not familiar to the neuroscience (e.g., Dale's law, brain's visual stream, brain region etc)
*   **Misleading Presentation of Core Methods:** Despite having detailed procedures in the Appendix, the minimal description in the main text creates unnecessary ambiguity regarding the source of the biological data (new recording vs. open-source) and the specifics of the complex feature visualization. This structure places an undue burden on the reader and reviewers and minimizes the necessary rigor for introducing novel electrophysiology data. **(Recommendation: Integrate a summary of all critical methods into the main body of the paper., even there are repeated section in methods and appendix)**. Moreover, figure 1A seems to describe the concept of method, how exactly monkey recording can reconstruct image while spliting exicatory or inhibitory is really hard to grasp.


2) Critical technical details are presented with insufficient formal rigor, making the analysis difficult to verify.

*   **Ablation Equation Ambiguity (Line 147):** The equation for ablation (e.g., using $\alpha$) lacks formal clarity. It is unclear if $w$ refers to layer-wise weights or the entire model, and whether the operation respects the sign of the weights or only their magnitude. Ambiguity regarding whether $\alpha$ can exceed 1 suggests a lack of precise definition for the weight's normalization or clipping.
*   **Undefined Notation ($L_{\infty}$):** The term $L_{\infty}$ (Line 138) must be explicitly defined (presumably the $\ell_{\infty}$ norm) as this basic notation should not be assumed or left to the Appendix for clarity in the main body.

3) The core premise of the cross-disciplinary comparison is built on a simplifying assumption that requires a more robust discussion.

*   **Structural Mismatch in Synapse Analogy:** The direct mapping of an ANN's signed weight to a biological synapse is structurally flawed. A standard ANN unit receives both positive and negative weighted inputs, whereas a biological neuron's *outgoing* connection typically adheres to Dale's Principle (single sign). The paper fails to rigorously address this significant **structural mismatch**, treating the *sign of the weight* as an equivalent proxy for the *sign of the synapse*. This omission weakens the fundamental premise that the observed feature segregation is truly analogous to biological circuit function and should be thoroughly discussed.

4) Lack of Justification for Sampling (Line 658)

- The authors' justification for data and class sampling is incomplete. The rationale for subsampling the **11 specific classes** for the analysis (Line 658) is not clearly articulated. Without a clear justification for selecting these particular classes, the generalizability of the reported results regarding feature segregation is questionable and may introduce selection bias.

**Questions:**

While the core intellectual efforts and the underlying experimental findings are commendable and demonstrate significant effort, the overall packaging and presentation of the manuscript detracts from its substantial contribution. The density of information, poor flow, and insufficient referencing in the main text make it unduly challenging for the reader to immediately grasp the rigor and context of the work. The authors are **strongly encouraged to significantly refine the narrative clarity and academic referencing** to appropriately honor the complexity and value of their research.

---

> ### Author Response · Authors · 2025-12-03
> **Reply to reviewer feedback**
>
> We thank the reviewer for acknowledging our multidisciplinary efforts to provide a fresh view to link mechanistic interpretability in AI and neuroscience.
> We address the points thoughtfully raised by the reviewer which focused on improving paper writing flow and clarity.
>
> ### Weakness 1 (clarity and methods descriptions)
> - Supplemented **Insufficient Referencing and Background**, especially in the introduction.
> - Rewrote "Misleading Presentation of Core Methods" by integrating summaries of relevant methods.
>
> ### Weakness 2 (clarity and technical details)
> - Corrected **Ablation Equation Ambiguity (Line 147)** including bounds on the ablation strength and added $L_\infty$ norm definition.
>
> ### Weakness 3 ( simplifying assumptions requires a more robust discussion)
> - We mention **Structural Mismatch in Synapse Analogy** to highlight loose analogy from the introduction and show that networks with better accuracy tend to be more sign consistent or Dale-like, which also correlates with network depth (Fig. 1). This without any explicit constraints, providing further validation for our loose analogy.
>
> ### Weakness 4 (class selection)
> - **Lack of Justification for Sampling (Line 658):** we used FastAI imagenette dataset as they selected 10 classes for small scale training. To reduce bias we validated other 100 classes sampled by k-means (Fig. 12, 13). Thus, we did not hand pick classes to avoid experimenter biases.
>
> #### Questions (**strongly encouraged to significantly refine the narrative clarity and academic referencing**)
> By addressing these points as well as those of other reviewers, we have fulfilled this compelling suggestion.

---

### Official Review · Reviewer_f5Cb · 2025-11-01

**Soundness:** 3
**Presentation:** 3
**Contribution:** 3
**Rating:** 6
**Confidence:** 3

**Summary:**

Across artificial and biological neural networks (macaque visual system) for vision, they show that positive weights emphasize objects, while negative weights encode context.

**Strengths:**

- Original study asking an important question about the role of negative vs positive synapses in the visual system and ANNs.
- Careful analyses which seem to mostly support the claims of the abstract.
- Clear writing for the most part.

**Weaknesses:**

- I am not entirely sure (but I may have missed some of the reasoning steps) that the claim of the abstract about the macaque visual system ("Our results demonstrate that both artificial and biological vision systems segregate features by weight sign: positive weights emphasize objects, negative weights encode context") is fully supported by the analyses provided. Could you please explain how the analyses support that claim?

- In the abstract, it is said: "Notably, some units closely approached Dale's law: the positively projecting units exhibited localized features, while the negatively projecting units showed larger, more dispersed features." How is this observation related to Dale's law, which states that a single neuron releases the same neurotransmitter at all of its synapses?

**Questions:**

- In order to test the emergence of (approximate) Dale's law in the artificial networks, one one need to run a statistical test to see if neurons tend to have more output connection weights of the same sign than expected by chance.

- Figures are a bit small. Consider enlarging in the final version.

---

> ### Author Response · Authors · 2025-12-03
> **Reply to reviewer comments**
>
> We thank the reviewer for the points, and hope that despite the situation we managed to better convey our message.
>
> ### Weakness 1 (support for abstract claim about macaque visual system)
> > “I am not entirely sure (but I may have missed some of the reasoning steps) that the claim of the abstract about the macaque visual system (…) is fully supported by the analyses provided. Could you please explain how the analyses support that claim?”
>
> There are two reasoning steps to support this conclusion. To improve clarity we edited the abstract to read: “The most Dale-like positively projecting units exhibited localized features, while the negatively projecting units showed larger, more dispersed features, suited to carrying contextual input. Consistent with this, clearing the background around each neuron's preferred feature enhanced its response, likely by reducing inhibitory drive, supporting inhibition as a contextual modulation of the excitatory feature.”
>
> 1)	Across neuron models, some input channels had consistently positive vs negative weights for more than 90% of the neurons. These are the Dale-like units (Fig. 23). Because they emerged without a Dale constraint, they provide good baselines for biological proxies of excitatory and inhibitory inputs. And their visualized features are consistent with a local/global or object/context organization.
>
> 2)	Then, in a model-free way, we cleared the background around visual features obtained in vivo and observed increased responses (Fig. 24). This confirms modulating the context of the original feature can increase neuron responses, which we interpret as a release of inhibitory drive activated by suboptimal context.
>
>
> ### Weakness 2 (approaching Dale's law)
> > In the abstract, it is said: "Notably, some units closely approached Dale's law: the positively projecting units exhibited localized features, while the negatively projecting units showed larger, more dispersed features." How is this observation related to Dale's law, which states that a single neuron releases the same neurotransmitter at all of its synapses?
>
> - We now explicitly add our Dale index to the methods. Briefly, the proportion of sign consistent outputs defines to what degree a unit follows Dale's law, relaxing the strict constraint to a continuous metric. Interestingly, we observe the coupling of weight sign to some visual features by filtering the most Dale-like output channels. And Fig. 1 shows that Dale index also relates to performance for different architectures further supporting utility of the continuous metric.
>
> #### Questions: testing significance of Dale's law
> > In order to test the emergence of (approximate) Dale's law in the artificial networks, one one need to run a statistical test to see if neurons tend to have more output connection weights of the same sign than expected by chance.
>
> - We selected units sending more than 90% of their outputs with same sign ensuring significance, and added the binomial test p-value of $p=5.09e^{-10}$ to the figure legend. We also included a new analysis showing that untrained networks do not show high Dale index, and training increases it (Fig. 1).
>
> #### Questions: small fonts
> - We already edited figures for improved clarity and font size, and will further do so when camera-ready version allows for an extra page.

---

### Author Response · Authors · 2025-12-03
**Overall reply**

We thank all reviewers for the points raised. And thank the AC for their critical role in keeping up ICLR scientific conduct standards.

All reviewers acknowledged the originality of the idea to explore sign role in shaping visual representations and our extensive efforts and biological experiments.

The main concerns were about (i) support for the object/context claim in macaque data, (ii) the relation to and significance of “Dale-like” units, (iii) clarity of methods and narrative, and (iv) the functional and biological interpretation of weight‑sign segregation. We addressed these as follows.

### 1. Biological validation and limitations
- We acknowledge the mismatch in expected data sizes for fitting models, since our models were validated in closed-loop we have limited time per session.
- Nevertheless, we show that perturbed models can recover features seen in closed‑loop in vivo feature visualization and can steer firing rates in the expected direction (though not with the fine control possible in ANNs; Figs. 7, 21).
- The background‑clearing experiment (Fig. 24) provides *model‑free* evidence that manipulating the surround/context of a preferred feature increases responses, compatible with a release‑from‑inhibition mechanism.
- We explicitly discuss limitations: most neurons are from V4/IT, fitted better by later layers, only 2 V1 neurons would be not get the best fit by using the penultimate layer.
- Dale's law is only loosely approached by the models in Fig. 23, and the feature analysis was key to motive our model-free background clearing.

### 2. Significance of Dale's law and its relation to ANNs
- Fig. 1 shows that even when CNNs are not required to follow biological Dale's law, for the same architecture deeper networks which achieve better performance also have more sign consistent organization (are Dale-like) in their output layers despite balancing total weight contribution per sign (Table 2).

### 3. Clarified methods, narrative, and referencing
In response to requests for clearer exposition:
- We expanded and better structured the introduction, adding the missing background and references and integrating concise summaries of the mechanistic‑interpretability and neuroscience methods we build on.
- We corrected the ablation equation, specified bounds on the ablation strength, and added the exact norm definition used.
- We clarified our class selection: we used the FastAI “imagenette” subset (10 classes) for small‑scale training, then verified that our findings generalize to 100 additional classes sampled by k‑means (Figs. 12–13), avoiding hand‑picked classes and experimenter bias.
- We also improved figure clarity and font sizes and will expand them further in the camera‑ready if an extra page is available.

### 4. Functional and biological interpretation of weight‑sign segregation.
- Figs. 14, 15 show that positive weights affect low frequency content in the preferred features consistent with low frequencies contributing more to objectness as detected by YOLO in Fig. 2.

- Fig. 16 shows that robustness increases not only the visual foreground/background appearance of features (Fig. 4, 5) but that this translates into more shape bias in shape/texture benchmark by Geirhos.

- While a definite benchmark to completely disambiguate shape/texture/background/foreground/local/global features is not available, we acknowledge the relation of these multiple features and report functional segregation between weights regardless of the ultimate interpretation.

### Conclusions
- We extensively edited the text for improve clarity and to add new figures, and extending existing figures.
- We hope our improved work on CNNs would be valuable to the field on their own, and that our biological data is mainly predicting concrete hypothesis for experimentalists in high level vision in monkeys and humans.

---

### Meta-Review · Area_Chair_7tVv · 2026-01-06

**Summary:**

Prior to the rebuttal period, two reviewers gave positive scores (6, 6) while two reviewers gave negative scores (2, 4). There was general concern among the reviewers that the experiments are insufficient to conclude that positive weights encode object-like features and negative weights encode background features. The reviewers primarily expressed concern that
- other factors (spatial structure, frequency context, etc.) can be conflated with the object versus background distinction
- weak evidence that the models can explain neural responses (poor variance explained, small datasets)

While the authors provided several new analyses to address these concerns, I think that ultimately this paper needs another round of review to ensure that the control experiments that the authors performed are sufficient to substantiate their claims.

**Reviewer Concerns:**

New analyses were performed to understand the effect of positive/negative weights on other factors that may be correlated with object/background distinctions. I believe the new analyses only partially address the conflation between other factors and the object/background distinction. For example, in Figure 14, the conclusion seems architecture-dependent: positive input ablations only appear to affect ResNet networks, not AlexNet or VGG networks. I'm also not sure the reviewers would have been convinced by Figure 16 that negative ablations more strongly disrupt texture encoding.

To address the concern that the networks are poor models of biological neurons, the authors point to their model free results. I believe the concern is still outstanding.

**Reviewer Scores:**

I believe that Reviewers f5Cb and udxK would have maintained their positive scores (6, 6). I believe that Reviewer kwmx would have increased their score from 4 to 6. I believe that Reviewer sJXU would have increased their score from 2 to 4. I think that while some concerns would remain, overall the reviewers would have recommended acceptance.

---

### Decision · Program_Chairs · 2026-01-26

Accept (Poster)